# SCALOR: GENERATIVE WORLD MODELS WITH SCALABLE OBJECT REPRESENTATIONS

**Jindong Jiang[†], Sepehr Janghorbani[†], Gerard de Melo & Sungjin Ahn**

Rutgers University

## ABSTRACT

Scalability in terms of object density in a scene is a primary challenge in unsupervised sequential object-oriented representation learning. Most of the previous models have been shown to work only on scenes with a few objects. In this paper, we propose SCALOR, a probabilistic generative world model for learning SCALable Object-oriented Representation of a video. With the proposed spatially-parallel attention and proposal-rejection mechanisms, SCALOR can deal with orders of magnitude larger numbers of objects compared to the previous state-of-the-art models. Additionally, we introduce a background module that allows SCALOR to model complex dynamic backgrounds as well as many foreground objects in the scene. We demonstrate that SCALOR can deal with crowded scenes containing up to a hundred objects while jointly modeling complex dynamic backgrounds. Importantly, SCALOR is the first unsupervised object representation model shown to work for natural scenes containing several tens of moving objects. https://sites.google.com/view/scalor/home

## 1 INTRODUCTION

Unsupervised structured representation learning for visual scenes is a key challenge in machine learning. When a scene is properly decomposed into meaningful entities such as foreground objects and background, we can benefit from numerous advantages of abstract symbolic representation. These include interpretability, sample efficiency, the ability of reasoning and causal inference, as well as compositionality and transferability for better generalization. In addition to symbols, another essential dimension is time. Objects, agents, and spaces all operate under the governance of time. Without accounting for temporal developments, it is often much harder if not impossible to discover certain relationships in a scene.

Among a few methods that have been proposed for unsupervised learning of object-oriented representation in temporal scenes, SQAIR (Kosiorek et al., 2018) is by far the most complete model. As a probabilistic temporal generative model, it can learn object-wise structured representation while modeling underlying stochastic temporal transitions in the observed data. Introducing the propagation-discovery model, SQAIR can also handle dynamic scenes where objects may disappear or be introduced in the middle of a sequence. Although SQAIR provides promising ideas and shows the potential of this important direction, a few key challenges remain, limiting its applicability merely to synthetic toy tasks that are far simpler than typical natural scenes.

The first and foremost limitation is scalability. Sequentially processing every object in an image, SQAIR has a fundamental limitation in scaling up to scenes with a large number of objects. As such, the state-of-the-art remains at the level of modeling videos containing only a few objects, such as MNIST digits, per image. Considering the complexity of typical natural scenes as well as the importance of scalable unsupervised object perception for applications such as self-driving systems, it is thus a challenge of the highest priority to scale robustly to scenes with a large number of objects. Scaling up the object-attention capacity is an important problem because it allows us to maximize the modern parallel computation that can maximize search capacity. This is contrary to humans, who can attend only to a few objects at a time in a time-consuming sequential manner. The

---

[†]Authors with equal contribution. Email addresses: {jindong.jiang, sepehr.janghorbani, gerard.demelo, sungjin.ahn}@rutgers.edu

AlphaGo system (Silver et al., 2017) is an example demonstrating the power of such parallel search (attention) beyond human capacity.

The second limitation is that previous models including SQAIR lack any form of background modeling and thus only cope with scenes without background, whereas natural scenes usually have a dynamic background. Thus, a temporal generative model that can deal with dynamic backgrounds along with many foreground objects is an important step toward natural video scene understanding.

In this paper, we propose a model called **SCAL**able Sequential **O**bject-Oriented **R**epresentation (SCALOR). SCALOR resolves the aforementioned key limitations and hence can model complex videos with several tens of moving objects along with dynamic backgrounds, eventually making the model applicable to natural videos. In SCALOR, we achieve scalability with respect to the object density by parallelizing both the propagation and discovery processes, reducing the time complexity of processing each image from $\mathcal{O}(N)$ to $\mathcal{O}(1)$, with $N$ being the number of objects in an image. We also observe that the sequential object processing in SQAIR, which is based on an RNN, not only increases the computation time but also deteriorates discovery performance. To this end, we propose a parallel discovery model with superior discovery capacity and performance. SCALOR can also be regarded as a generative tracking model since it not only detects object trajectories but is also able to predict trajectories into the future. In our experiments, we demonstrate that SCALOR can model videos with nearly one hundred moving objects along with a dynamic background on synthetic datasets. Furthermore, we showcase the ability of SCALOR to operate on natural-scene videos containing tens of objects with a dynamic background.

The contributions of this work are: (i) We propose the SCALOR model, which significantly improves (two orders of magnitude) the scalability in terms of object density. It is applicable to nearly a hundred objects while providing more efficient computation time than SQAIR. (ii) We propose parallelizing the propagation–discovery process by introducing the propose–reject model, reducing the time complexity to $\mathcal{O}(1)$. (iii) SCALOR can model scenes with a dynamic background. (iv) SCALOR is the first probabilistic model demonstrating its working on a significantly more complex task, i.e., natural scenes containing tens of objects as well as background.

## 2 PRELIMINARIES: SEQUENTIAL ATTEND INFER REPEAT (SQAIR)

SQAIR models a sequence of images $\mathbf{x} = \mathbf{x}_{1:T}$ by assuming that observation $\mathbf{x}_t$ at time $t$ is generated from a set of object latent variables $\mathbf{z}_t^{\mathcal{O}} = \{\mathbf{z}_{t,n}\}_{n \in \mathcal{O}_t}$ with $\mathcal{O}_t$ the set of objects present at time $t$. Latent variable $\mathbf{z}_{t,n}$ corresponding to object $n$ consists of three factors $(z_{t,n}^{\text{pres}}, \mathbf{z}_{t,n}^{\text{where}}, \mathbf{z}_{t,n}^{\text{what}})$, which represent the existence, pose, and appearance of the object, respectively. SQAIR also assumes that an object can disappear or be introduced in the middle of a sequence. To model this, it introduces the propagation–discovery model. In propagation, a subset of currently existing objects is propagated to the next time-step and those not propagated (e.g., moved out of the scene) are deleted. In discovery, after deciding how many objects $D_t$ will be discovered, $D_t$ objects are newly introduced into the scene. Combining the propagated set $\mathcal{P}_t$ and discovered set $\mathcal{D}_t$, we obtain the set of currently existing objects $\mathcal{O}_t$. The overall process can be formalized as:

$$p(\mathbf{x}_{1:T}, \mathbf{z}_{1:T}^{\text{fg}}, D_{1:T}) = p(D_1, \mathbf{z}_1^{\mathcal{D}}) \prod_{t=2}^{T} p(\mathbf{x}_t | \mathbf{z}_t^{\text{fg}}) \, p(D_t, \mathbf{z}_t^{\mathcal{D}} | \mathbf{z}_t^{\mathcal{P}}) \, p(\mathbf{z}_t^{\mathcal{P}} | \mathbf{z}_{t-1}^{\text{fg}}) \,. \tag{1}$$

For SQAIR, we use $\mathbf{z}_t^{\text{fg}}$ ("fg" standing for foreground) to denote $\mathbf{z}_t^{\mathcal{O}}$ because SQAIR does not have any latent variables for background. Due to the intractable posterior, SQAIR is trained through variational inference with the following posterior approximation:

$$q(D_{1:T}, \mathbf{z}_{1:T}^{\text{fg}} | \mathbf{x}_{1:T}) = \prod_{t=1}^{T} q(D_t, \mathbf{z}_t^{\mathcal{D}} | \mathbf{x}_t, \mathbf{z}_t^{\mathcal{P}}) \prod_{n \in \mathcal{O}_{t-1}} q(\mathbf{z}_{t,n} | \mathbf{z}_{t-1,n}, \mathbf{x}_{\leq t}) \,. \tag{2}$$

SQAIR is trained using the importance-weighted autoencoder (IWAE) objective (Burda et al., 2015). The VIMCO estimator (Mnih & Rezende, 2016) is used to backpropagate through the discrete random variables while using the reparameterization trick (Kingma & Welling, 2013; Williams, 1992) for continuous variables. SQAIR has two main limitations in terms of scalability. First, for propagation, SQAIR relies on an RNN, which sequentially processes each object by conditioning on previously processed objects. Second, the discovery is also sequential because it uses RNN-based

discovery based on AIR (Eslami et al., 2016). Consequently, SQAIR has a time complexity of $O(|\mathcal{O}_t|)$ per step $t$. In Crawford & Pineau (2019b), the authors demonstrated that this sequential discovery can easily fail beyond the scale of a few objects. Moreover, SQAIR lacks any model for the background and its temporal transitions, which is important in modeling natural scenes.

## 3    SCALOR

In this section, we describe the proposed model, SCALOR. We first describe the generative process along with the proposal-rejection mechanism, which is designed to prevent propagation collapse, and then the inference process and learning.

### 3.1    GENERATIVE PROCESS

SCALOR assumes that an image $\mathbf{x}_t$ is generated by background latent $\mathbf{z}_t^{\mathrm{bg}}$ and foreground latent $\mathbf{z}_t^{\mathrm{fg}}$. The foreground is further factorized into a set of object representations $\mathbf{z}_t^{\mathrm{fg}} = \{\mathbf{z}_{t,n}\}_{n \in \mathcal{O}_t}$. In SCALOR, we represent an object by $\mathbf{z}_{t,n} = (z_{t,n}^{\mathrm{pres}}, \mathbf{z}_{t,n}^{\mathrm{where}}, \mathbf{z}_{t,n}^{\mathrm{what}})$ similarly to SQAIR. The appearance representation $\mathbf{z}_{t,n}^{\mathrm{what}}$ is a continuous vector representation, and $\mathbf{z}_{t,n}^{\mathrm{where}}$ is further decomposed into the center position $\mathbf{z}_{t,n}^{\mathrm{pos}}$, scale $\mathbf{z}_{t,n}^{\mathrm{scale}}$, and depth $z_{t,n}^{\mathrm{depth}}$. The depth representation, which is missing in SQAIR, represents the relative depth between objects from the camera viewpoint. This depth modeling helps deal with object occlusion. The foreground mask $\mathbf{m}_{t,n}$ obtained from $\mathbf{z}_{t,n}^{\mathrm{what}}$ is used to distinguish background and foreground. We adopt the propagation–discovery model from SQAIR, but improve it in such a way to resolve the scalability problem. The generative process of SCALOR can be written as:

$$p(\mathbf{x}_{1:T}, \mathbf{z}_{1:T}) = p(\mathbf{z}_1^{\mathcal{D}})(\mathbf{z}_1^{\mathrm{bg}}) \prod_{t=2}^{T} \underbrace{p(\mathbf{x}_t|\mathbf{z}_t)}_{\text{rendering}} \underbrace{p(\mathbf{z}_t^{\mathrm{bg}}|\mathbf{z}_{<t}^{\mathrm{bg}}, \mathbf{z}_t^{\mathrm{fg}})}_{\text{background transition}} \underbrace{p(\mathbf{z}_t^{\mathcal{D}}|\mathbf{z}_t^{\mathcal{P}})}_{\text{discovery}} \underbrace{p(\mathbf{z}_t^{\mathcal{P}}|\mathbf{z}_{<t})}_{\text{propagation}}, \qquad (3)$$

where $\mathbf{z}_t = (\mathbf{z}_t^{\mathrm{bg}}, \mathbf{z}_t^{\mathrm{fg}})$. As shown, the generation process is decomposed into four modules: (i) propagation, (ii) discovery, (iii) background transition, and (iv) rendering.

**Propagation.** The propagation in SCALOR is modeled as follows:

$$p(\mathbf{z}_t^{\mathcal{P}}|\mathbf{z}_{<t}) = \prod_{n \in \mathcal{O}_t} p(z_{t,n}^{\mathrm{pres}}|\mathbf{z}_{<t,n}) \left\{ p(\mathbf{z}_{t,n}^{\mathrm{where}}|\mathbf{z}_{<t,n}) \, p(\mathbf{z}_{t,n}^{\mathrm{what}}|\mathbf{z}_{<t,n}) \right\}^{z_{t,n}^{\mathrm{pres}}}, \qquad (4)$$

where $p(z_{t,n}^{\mathrm{pres}}|\mathbf{z}_{<t,n})$ is a Bernoulli distribution with parameter $\beta_{t,n}$. The distributions of "what" and "where" are defined only when the object is propagated. To implement this, for each object $n$ we assign an object-tracker RNN denoted by its hidden state $\mathbf{h}_{t,n}$. The RNN is updated by input $\mathbf{z}_{t,n}$ for all $t$ where the object $n$ is present in the scene. The parameter $\beta_{t,n}$ is obtained as $\beta_{t,n} = f_{\mathrm{mlp}}(\mathbf{h}_{t,n})$. If $z_{t,n}^{\mathrm{pres}} = 0$, the object $n$ is not propagated and the tracker RNN is deleted. Importantly, unlike the RNN-based sequential propagation in SQAIR, the propagation in SCALOR is fully parallel.

**Discovery by Proposal-Rejection.** The main contribution in making our model scalable with respect to the the number of objects is our new discovery model that consists of two phases: *proposal* and *rejection*. In the proposal phase, we assume that the target image can be covered by $H \times W$ latent grid cells, and we propose an object latent variable $\tilde{\mathbf{z}}_{t,h,w}$ per grid cell. This proposal phase can be written as:

$$p(\tilde{\mathbf{z}}_t^{\mathcal{D}}|\mathbf{z}_t^{\mathcal{P}}) = \prod_{h,w=1}^{HW} p(\tilde{\mathbf{z}}_{t,h,w}^{\mathcal{D}}|\mathbf{z}_t^{\mathcal{P}}) = \prod_{h,w=1}^{HW} p(\tilde{z}_{t,h,w}^{\mathrm{pres}}|\mathbf{z}_t^{\mathcal{P}}) \left\{ p(\tilde{\mathbf{z}}_{t,h,w}^{\mathrm{where}}|\mathbf{z}_t^{\mathcal{P}}) \, p(\tilde{\mathbf{z}}_{t,h,w}^{\mathrm{what}}|\mathbf{z}_t^{\mathcal{P}}) \right\}^{\tilde{z}_{t,h,w}^{\mathrm{pres}}}. \qquad (5)$$

In the rejection phase, our goal is to reject some of the proposed objects if a proposed object largely overlaps with a propagated object. We realize this by using the mask variable $\mathbf{m}_{t,n}$. Specifically, if the overlapping area between the mask of a proposed object and that of a propagated object is over a threshold $\tau$, we reject the proposed object. This procedure can be described as (i) proposal: $\tilde{\mathbf{z}}_t^{\mathcal{P}} \sim p(\tilde{\mathbf{z}}_t^{\mathcal{P}}|\mathbf{z}_t^{\mathcal{P}})$ and (ii) accept-reject: $\mathbf{z}_t^{\mathcal{D}} = f_{\text{accept-reject}}(\tilde{\mathbf{z}}_t^{\mathcal{D}}, \mathbf{z}_t^{\mathcal{P}}, \tau)$. In this way, the final discovery set $\mathbf{z}_t^{\mathcal{D}}$ is always a subset of the proposal set $\tilde{\mathbf{z}}_t^{\mathcal{D}}$, i.e., $\mathbf{z}_t^{\mathcal{D}} \subseteq \tilde{\mathbf{z}}_t^{\mathcal{D}}$. Although we use a deterministic function for the rejection, it can be a design choice to implement this as a stochastic decision. While one rationale behind this design is to reflect an inductive bias of a Gestalt

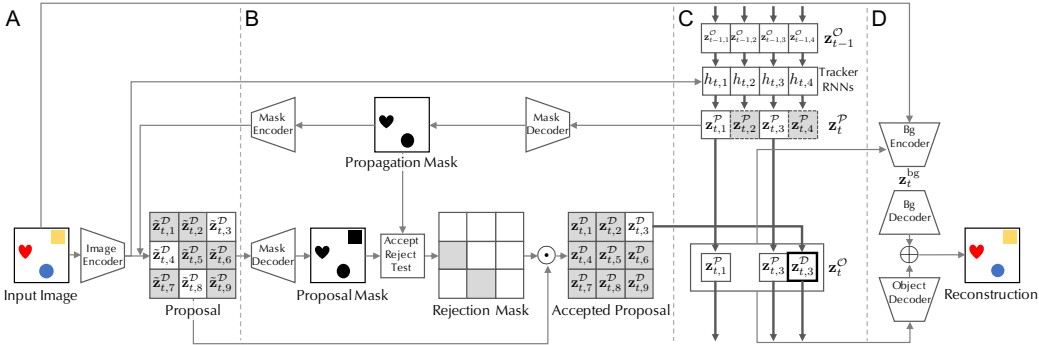

**Figure 1:** SCALOR inference procedure: (A) Proposal, (B) Accept-Reject, (C) Propagation, (D) Background Module and Rendering Process. (A) The proposal module takes the input image and propagation mask and combines them to make the proposal representation. (B) From the proposal representation, the proposal mask is generated, and then compared to the propagation mask to make an accept-reject decision. Only the accepted proposals are considered as discovered objects. (C) The tracker RNNs decide what and where to propagate after looking at the input image. The gray boxes represent what is not propagated. (D) Given inferred foreground objects and the input image, the background module infers the background representation. The rendering process combines the foreground and background representations according to the foreground mask assignment

principle saying that two objects cannot coexist in the same position, we shall also see later further reasons as to why this design is effective. The final discovery model can then be written as: $p(\mathbf{z}_t^{\mathcal{D}}|\mathbf{z}_t^{\mathcal{P}}) = p(\tilde{\mathbf{z}}_t^{\mathcal{D}}|\mathbf{z}_t^{\mathcal{P}}) \prod_{h,w=1}^{HW} p(\mathbf{z}_{t,h,w}^{\mathcal{D}}|\mathbf{z}_t^{\mathcal{P}}, \tilde{\mathbf{z}}_t^{\mathcal{D}}, \tau)$, where the acceptance model is:

$$p(\mathbf{z}_{t,h,w}^{\mathcal{D}}|\mathbf{z}_t^{P}, \tilde{\mathbf{z}}_t^{\mathcal{D}}, \tau) = f(z_{t,h,w}^{\text{accept}}|\mathbf{z}_t^{P}, \tilde{\mathbf{z}}_t^{\mathcal{D}}, \tau) p(\tilde{\mathbf{z}}_{t,h,w}^{\text{where}})^{z_{t,h,w}^{\text{accept}}} p(\tilde{\mathbf{z}}_{t,h,w}^{\text{what}})^{z_{t,h,w}^{\text{accept}}} . \tag{6}$$

**Background Transition.** Unlike SQAIR, SCALOR is endowed with a background model. The background image is encoded into a $D$-dimension continuous vector $\mathbf{z}_t^{\text{bg}}$ from the background transition $p(\mathbf{z}_t^{\text{bg}}|\mathbf{z}_{<t}^{\text{bg}}, \mathbf{z}_t^{\text{fg}})$. The background RNN encodes the temporal transition of background images.

**Rendering.** The implementation of the rendering process is the same as in SPAIR (Crawford & Pineau, 2019b), except that we process the objects in parallel. Implementation details are shown in Appendix A.

### 3.2 LEARNING AND INFERENCE

Due to the intractability of the true posterior distribution $p(\mathbf{z}_{1:T}|\mathbf{x}_{1:T})$, we train our model using variational inference with the following posterior approximation:

$$q(\mathbf{z}_{1:T}|\mathbf{x}_{1:T}) = \prod_{t=1}^{T} q(\mathbf{z}_t|\mathbf{z}_{<t}, \mathbf{x}_{\leq t}) = \prod_{t=1}^{T} q(\mathbf{z}_t^{\text{bg}}|\mathbf{z}_t^{\text{fg}}, \mathbf{x}_t) \, q(\mathbf{z}_t^{\mathcal{D}}|\mathbf{z}_t^{\mathcal{P}}, \mathbf{x}_{\leq t}) \, q(\mathbf{z}_t^{\mathcal{P}}|\mathbf{z}_{<t}, \mathbf{x}_{\leq t}) . \tag{7}$$

**Posterior Propagation.** $q(\mathbf{z}_t^{\mathcal{P}}|\mathbf{z}_{<t}, \mathbf{x}_{\leq t})$ is similar to the propagation in generation, except that we now provide observation $\mathbf{x}_{\leq t}$ through an RNN encoding. Here, the propagation for each object $n$ is done by $q(\mathbf{z}_{t,n}|\mathbf{z}_{<t,n}, \mathbf{x}_{\leq t})$ using attention $\mathbf{a}_{t,n} = f_{\text{att}}(\mathbf{x}_{\leq t})$ on the feature map for object $n$. To compute the attention, we use the previous position $\mathbf{z}_{t-1,n}^{\text{pos}}$ as the center position and extract half the width and height of the convolutional feature map using bilinear interpolation. This attention mechanism is motivated by the observation that only part of the image contains information for tracking an object and an inductive bias that objects cannot move a large distance within a short time span (i.e., objects do not teleport).

**Posterior Discovery.** The posterior discovery also consists of proposal and rejection phases. The main difference is that we now compute the proposal in *spatially-parallel* manner by conditioning on the observations $\mathbf{x}_{\leq t}$, i.e., $q(\tilde{\mathbf{z}}_t^{\mathcal{D}}|\mathbf{z}_t^{\mathcal{P}}, \mathbf{x}_{\leq t}) = \prod_{h,w=1}^{HW} q(\tilde{\mathbf{z}}_{t,h,w}^{\mathcal{D}}|\mathbf{z}_t^{\mathcal{P}}, \mathbf{x}_{\leq t})$. Here, the observation $\mathbf{x}_{\leq t}$ is encoded into the feature map of dimensionality $H \times W \times D$ using a Convolutional LSTM (Xingjian et al., 2015). Then, from each feature we obtain $\tilde{\mathbf{z}}_{t,h,w}^{\mathcal{D}}$. Importantly, this is done over all the feature cells $h, w$ in parallel. A similar approach is used in SPAIR (Crawford & Pineau, 2019b), but it infers the object latent representations sequentially and thus is difficult to scale to a large number

of objects (Lin et al., 2020). Even if this spatially-parallel proposal plays a key role in making our model scalable, we also observe another challenge due to this high capacity of the discovery module. The problem is that the discovery module tends to dominate the propagation module and thus most of the objects in an image are explained by the discovery module, i.e., objects are *re*discovered at every time-step while nothing is propagated. We call this problem *propagation collapse*.

Why would the model tend to explain an image through discovery while suppressing propagation? First, the model does not care where—either from discovery or propagation—an object is sourced from as long as it can make an accurate reconstruction. Second, the propagation step performs a much harder task than the discovery. For the propagation to properly predict, tracker $n$ needs to learn to *find the matching object* from an image containing many objects. Although the propagation attention plays an important role in balancing the discovery and propagation, we found that it does not eliminate the problem of re-discovery, and without rejection, its effectiveness varies across different experiment settings. On the contrary, the discovery module does not need to solve such a difficult association problem because it only performs *local* image-to-latents encoding without associating latents of the previous time-step. Therefore, it is much easier for the discovery encoder to produce latents that are more accurate than those inferred from propagation. If we limit the capacity of the discovery module and sequentially process objects like in SQAIR, we may mitigate this problem because the propagation module is naturally encouraged to explain what this weakened discovery module cannot. This approach, however, cannot scale.

We employ two techniques to resolve the aforementioned problem. First, we simply bias the initial network parameter so that it has a high propagation probability at the beginning of the training. This helps the model prefer to explain the observation first through propagation. The second technique is our proposal-rejection mechanism, which is implemented the same way as in the generation process. This prevents the discovery model from redundantly explaining what is already explained by the propagation module. The posterior for the discovery model can be written as:

$$q(\mathbf{z}_t^{\mathcal{D}}|\mathbf{z}_t^{\mathcal{P}}, \mathbf{x}_{\leq t}) = q(\tilde{\mathbf{z}}_t^{\mathcal{D}}|\mathbf{z}_t^{\mathcal{P}}, \mathbf{x}_{\leq t}) \prod_{h,w=1}^{HW} p_{\text{accept}}(\mathbf{z}_{t,h,w}^{\mathcal{D}}|\mathbf{z}_t^{\mathcal{P}}, \tilde{\mathbf{z}}_t^{\mathcal{D}}) , \tag{8}$$

where the acceptance model is $p_{\text{accept}}(\mathbf{z}_{t,h,w}^{\mathcal{D}}|\mathbf{z}_t^{\mathcal{P}}, \tilde{\mathbf{z}}_t^{\mathcal{D}}) = p(z_{t,h,w}^{\text{pres}}|\mathbf{z}_t^{\mathcal{P}}, \tilde{\mathbf{z}}_t^{\mathcal{D}})(p(\tilde{\mathbf{z}}_{t,h,w}^{\text{where}})p(\tilde{\mathbf{z}}_{t,h,w}^{\text{what}}))^{z_{t,h,w}^{\text{pres}}}$.

**Posterior Background.** The posterior of the background $q(\mathbf{z}_t^{\text{bg}}|\mathbf{z}_t^{\text{fg}}, \mathbf{x}_t)$ is conditioned on the input image and currently existing objects. Here, we provide the foreground latents so that the remaining parts in the image can be explained by the background module.

**Training.** We train our model by maximizing the following evidence lower bound $\mathcal{L}(\theta, \phi) =$

$$\sum_{t=1}^{T} \mathbb{E}_{q_\phi(\mathbf{z}_{<t}|\mathbf{x}_{<t})} \left[ \mathbb{E}_{q_\phi(\mathbf{z}_t|\mathbf{z}_{<t}, \mathbf{x}_{\leq t})} \left[ \log p_\theta(\mathbf{x}_t|\mathbf{z}_t) \right] - \mathbb{KL} \left[ q_\phi(\mathbf{z}_t|\mathbf{z}_{<t}, \mathbf{x}_{\leq t}) \parallel p_\theta(\mathbf{z}_t|\mathbf{z}_{<t}) \right] \right] . \tag{9}$$

We use the reparameterization trick (Williams, 1992; Kingma & Welling, 2013) for continuous random variables such as $\mathbf{z}^{\text{what}}$, and the Gumbel-Softmax trick (Jang et al., 2016) for discrete variables such as $z^{\text{pres}}$. We found that our proposed model works well and robustly with these simpler training methods than what is used in SQAIR, i.e., VIMCO and IWAE.

## 4 RELATED WORK

Different approaches have been taken to tackle the problem of unsupervised scene representation learning. Object-oriented models such as AIR (Eslami et al., 2016) and SPAIR (Crawford & Pineau, 2019b) decompose scenes into latent variables representing the appearance, position, and size of the underlying objects. While AIR makes use of a recurrent neural network, SPAIR applies spatially invariant attention to extract local feature maps. Although the latter provides better scalability than AIR, it is still limited as it performs sequential inference on objects. SQAIR (Kosiorek et al., 2018), discussed in Section 2, extends the ideas proposed in AIR to temporal sequences. On the other hand, scene-mixture models (Greff et al., 2017; Van Steenkiste et al., 2018; Burgess et al., 2019; Greff et al., 2019; Engelcke et al., 2019) decompose scenes into a collection of components, each being a full-image level representation. Although such models allow decomposition of the input image into components, they are not object-wise disentangled as multiple objects can be in the same

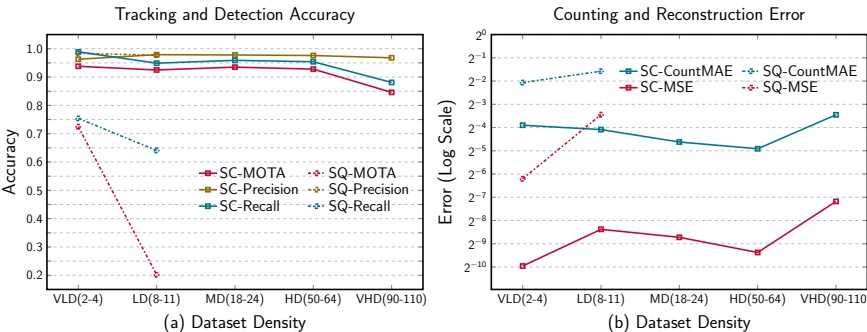

**Figure 2:** Quantitative result showing superior performance of SCALOR (SC) compared to SQAIR (SQ). (a) Tracking Accuracy (b) Object Count and Reconstruction Error

component. Furthermore the obtained representation does not contain explicit interpretable features like position and scale, etc. SPACE (Lin et al., 2020) combines both of the above approaches by using object detection for foreground and mixture decomposition for background. It improves upon SPAIR by parallelizing the latent inference process.

DDPAE (Hsieh et al., 2018) is another object-oriented sequential generative model that models each object with an appearance and a position vector. The model assumes the appearance of an object to be fixed, and thus shares the content vector across different time-steps. NEM (Greff et al., 2017) and RNEM (Van Steenkiste et al., 2018) introduce a spatial mixture model to disentangle the scene into multiple components representing each entity. Since each component generates a full scene image, the latent representations are not interpretable. Tracking-By-Animation (He et al., 2019) introduces a deterministic model to tackle the task of object tracking in an unsupervised fashion. Furthermore, there is a substantial amount of work on object tracking from the computer vision community using the same "bounding box"-representation approach proposed in SCALOR (Kosiorek et al., 2017; Ning et al., 2017; Nam & Han, 2016; Tao et al., 2016), but such methods use provided labels to tackle the problem of object tracking and thus are usually fully or semi-supervised and not probabilistic object-oriented models.

We also note that Crawford & Pineau (2019a) has independently and concurrently developed a similar architecture to ours. This model also emphasizes the scalability problem with a similar idea motivated by parallelizing SPAIR and extending it to sequential modeling. The main differences are the usage of the proposal-rejection mechanism and the background modeling that make our model work on complex natural scenes.

## 5 EXPERIMENTS

In this section, we describe the experiments conducted to empirically evaluate the performance of SCALOR. We propose two tasks, (i) synthetic MNIST/dSprites shapes and (ii) natural-scene CCTV footage of walking pedestrians. We will show SCALOR's abilities to detect and track objects, to generate future trajectories, and to generalize to unseen settings. Furthermore, we provide a quantitative comparison to state-of-the-art baselines.

### 5.1 TASK 1: LARGE-SCALE MNIST AND DSPRITES SHAPES

We first evaluate our model on datasets of moving dSprites shapes as well as moving MNIST digits. In all experiments, the image sequence covers a $64 \times 64$ partial view of the center of the whole environment. Therefore, while there is a fixed number of objects in the environment at each time-step, only a subset of them are visible in the observed image. The environment size is customized for each setting in a way that objects can conveniently move out of the viewpoint completely and re-enter within a few time-steps. We test on five different scale settings. In each setting, the number of objects in each trajectory is sampled uniformly from an interval [*min*, *max*]. Each *scale setting* is specified with a triplet (*min*, *avg*, *max*), where *min* and *max* are as mentioned and *avg* represents the average number of visible objects in the trajectories in that setting. The five settings are referred to as Very Low Density (VLD) [(2, 2.9, 4)], Low Density (LD) [(8, 8, 11)], Medium Density (MD) [(18, 20, 24)], High Density (HD) [(50, 55, 64)] and Very High Density (VHD) [(90, 90, 110)]. For

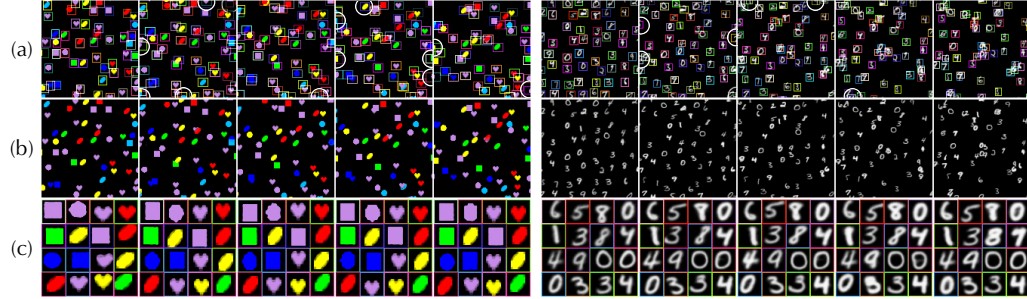

**Figure 3:** Qualitative results of SCALOR for Moving dSprites and Moving MNIST (HD) tasks: a) Inferred bounding boxes superimposed on the original image sequence. White circles indicate discovery at that timestep, b) Reconstructed sequence, c) Per-object reconstruction

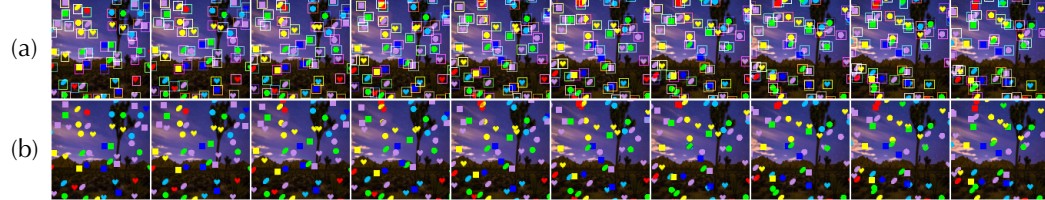

**Figure 4:** Qualitative samples of tracking on Moving dSprites task with dynamic background: a) Original image sequence with inferred bounding boxes, b) Reconstructed sequence

example, in the MD setting, there are always between 18 to 25 objects in the overall environment, while only about 20 of them are visible on average in each frame.

**Experiment 1 - Tracking Performance.** This experiment evaluates the tracking performance in an environment without background. We use the following tracking metrics: Multi-Object Tracking Accuracy (MOTA), precision-recall of the inferred bounding boxes (Bernardin & Stiefelhagen, 2008), reconstruction mean squared error, and normalized counting mean absolute error (Count-MAE). CountMAE measures the difference between the number of predicted objects and the actual number of objects, normalized by the latter. MOTA measures the tracking accuracy and consistency of the bounding boxes. Precision-recall measures the number of false-positive and negatives, regardless of the associated IDs. For computing the MOT metrics, we choose the Euclidean distance threshold to be twice the actual object size in each setting. Figure 2 shows the quantitative results of SCALOR compared to baselines. More detailed quantitative results are given in Appendix B.

We compare the performance of the proposed model with SQAIR in the VLD setting for both MNIST and dSprites datasets, and LD setting for dSprites. Note that we were not able to make SQAIR work on other settings due to the high object density. As shown in Figure 2, SCALOR outperforms SQAIR in all these settings, obtaining significantly higher accuracy and recall. SQAIR either misses some objects or misidentifies distinct objects as one. In addition, SCALOR has lower values of CountMAE in comparison to SQAIR, showing that SCALOR can infer the number of objects in the scene more accurately. Furthermore, we observe that increasing the number of objects in the scene does not significantly impede SCALOR's tracking ability, which demonstrates the strength of SCALOR when applied to images with a high number of objects. SCALOR can achieve relatively high precision-recall even for scenes containing about 100 objects. Note that in the VHD case, the number of objects (about 90) in the first time-step exceeds the number of detection grid cells (8×8) the discovery model has. Thus, the model can only detect up to 64 objects at the first time-step, and detects the rest in the following time-steps. This results in lower performance on the tracking and detection accuracy in the VHD case. This is demonstrated in Figure 10 in Appendix C.

Figure 3 demonstrates the qualitative performance of SCALOR on dSprites and MNIST (HD). To clarify tracking consistency, bounding boxes with distinct ids are represented by distinct colors. Discovered objects are emphasized by white circles. As shown in Figure 3(a), the discovery module of SCALOR can identify newly introduced objects and put them in the propagation list while the propagation module keeps tracking existing objects. Figure 3(c) shows object-wise rendering of inferred $\mathbf{z}^{\text{what}}$ latent variables. For clear visualization, object-wise rendering is shown only for a

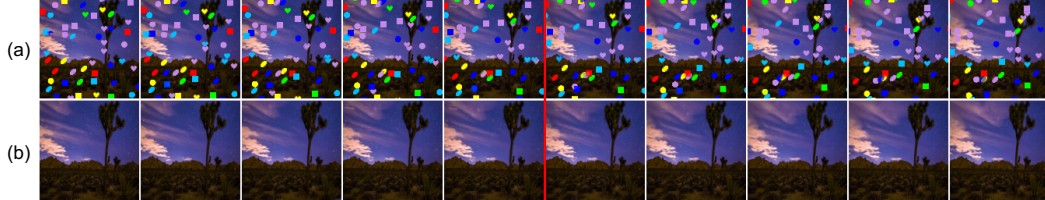

**Figure 5:** Conditional generation. The first 5 frames are provided to the model while the next 5 are generated. The red line indicates where generation starts from. a) Image reconstruction/generation. b)Dynamic background inference/generation

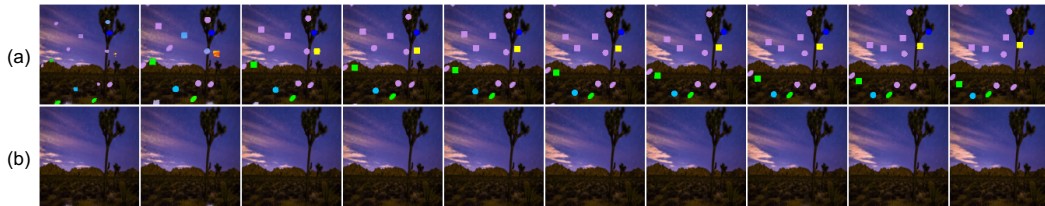

**Figure 6:** Generation from scratch. Objects are sampled from discovery prior and propagated in the next time frames. a) Generated sequence. b) Generated dynamic background

subset of objects present in all the time-steps. SCALOR infers consistent object-wise representation even when objects are largely occluded.

**Experiment 2 - Dynamic Background.** Another interesting yet challenging setting is the presence of a dynamic background. As we can see in Figure 4, SCALOR can decompose the video sequence to a set of foreground objects as well as a dynamic background. While the background module can model the background dynamics, it can also model the dynamics of the whole environment including the moving objects. This usually brings competition between background module and foreground module on explaining the foreground part. However, the background module and foreground module cooperate properly in SCALOR. We believe this is because of our modeling that the background module obtains information from the foreground part by conditioning on the foreground latent variables. Table 1 also provides the tracking performance of the model for this setting to show how complex images affect the tracking quality. We observe that it achieves comparable performance to the default no-background setting. These results are provided in Table 1 under "SCALOR – BG".

**Experiment 3 - Future Time Prediction/Generation.** This section showcases the generation ability of SCALOR via two experiments. The first experiment is aimed at showing conditional generation. Here, the model is provided with the first 5 frames, and then tested to generate the next 5 frames. Latent variables at each time-step are sampled independently from the prior distribution of the propagation step conditioned on latent variables from previous time-step. Because the focus is on conditional generation of background and objects, discovery prior of $\mathbf{z}^{\text{pres}}$ is manually set to zero. Figure 5 shows a generated sample. The first 5 frames (before the red line) correspond to reconstruction while the next 5 frames represent conditional generation[1]. As we can see in Figure 5, SCALOR can generate dynamic background and object trajectories that are reasonably consistent.

The second experiment showcases video generation from scratch. In this setting, the model generates all latent variables from the discovery prior at the first time-step. For the following time-steps, latent variables are sampled by conditioning on the latent variables of the previous time-steps. The object generation probability for each grid cell at the first time-step, i.e. $\mathbf{z}^{\text{pres}}_{1,h,w}$, is set to 0.2. Figure 6 shows one generated sample. At the first time-step where all variables are sampled from a fixed prior, the background is generated reasonably, but the objects are generated only partially. This behavior is expected because newly introduced objects are mostly partially observed at image boundaries. This makes the inference network learn a $\mathbf{z}^{\text{what}}$ representation corresponding to the partial views. Furthermore, since the $\mathbf{z}^{\text{what}}$ prior is a standard Gaussian, it is hard to model the multi-modal object appearance. Yet, it is interesting to see that the conditional generation of each object's $\mathbf{z}^{\text{what}}$ gradually converges to a consistent complete view of an object. This demonstrates that in our model,

---

[1]Examples with longer duration are in the project website

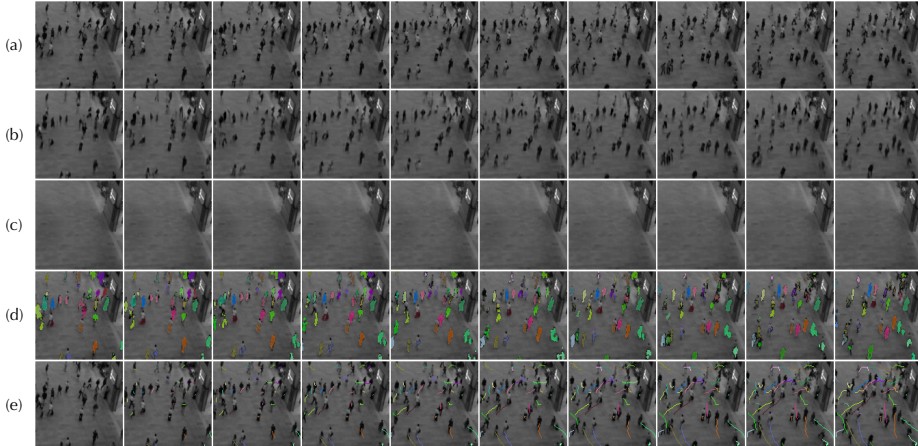

**Figure 7:** SCALOR's performance on Grand Central Station Dataset. (a) input sequence, (b) overall reconstruction of objects and background, (c) reconstruction of the extracted background, (d) segmentation for each object, colors indicate tracking ID, (e) extracted object trajectories

the distribution of $\mathbf{z}^{\text{what}}_{t \geq 2}$ in propagation can actually achieve multi-modality when conditioned on $\mathbf{z}^{\text{what}}_1$ from the first time-step. Furthermore, it shows that the propagation prior network is capable of maintaining a consistent representation of the object along the sequence as is reflected in the dataset. Therefore, SCALOR is capable of generating objects with independent appearance and behavior.

## 5.2 Task 2: Real-World Scenario

This section considers the performance of SCALOR on real-world natural video, which previous models have not been able to handle due to the scalability issue and lack of a background model. Compared to synthetic data, the challenges in this setting are significantly more difficult. Specifically, we consider the Crowded Grand Central Station dataset (Zhou et al., 2012), which was collected from CCTV cameras at New York City's Grand Central Station. Due to the complex pedestrian behavior, the density of the dataset can be considered a mix of LD and HD.

Figure 7 shows the tracking result on a sample sequence. We can see that SCALOR performs reasonably well on this pedestrian tracking dataset by maintaining consistent temporal trajectories. As shown in Figure 7(c), the background module infers the background component and reconstructs the extracted background correctly. As for object detection, SCALOR succeeds in accurate pedestrian detection and tracking. Furthermore, the foreground mask produced by $\mathbf{z}^{\text{what}}$ provides the segmentation of each individual pedestrian, as shown in Figure 7(d). We draw tracking trajectories in Figure 7(e) for each pedestrian in the natural scene. Trajectories in different colors correspond to different pedestrian ids inferred by the identity of the latent variable of each object. Additional figures and the dataset details are provided in Appendix D.

Figure 14 in Appendix D shows the conditional generation. The first 5 frames are the inference reconstruction while the last 5 frames are model generation. Starting from the 6th frame, the latent transition of the propagation trackers is modeled by the sequential prior network. In the generation process, a different prior of $z^{\text{pres}}$ is introduced in the discovery module to introduce new objects emerging in the scene (see Appendix E for more details). As shown in the figure, the model tends to generate movement aligned with the previous frames. This also applies to newly generated objects from the discovery phase as shown in Figure 14(f). Although the trajectories are consistent, the appearance of generated objects fails to maintain its consistency across different time frames. Noticeably, in the generated sequence the segmentation mask for each object tends to deform into a different shape. This may stem from imperfections during the inference that makes the learning of appearance transition difficult. Additional figures for generation are provided in Appendix D.

The ground truth trajectories of the Grand Central Station Dataset were not available when this experiment was conducted. We instead use negative log-likelihood (NLL) to compare our model with two baselines, a sequential VAE and a Recurrent Latent Variable Model (VRNN). The sequential VAE has one latent variable $\mathbf{z}$ of dimensionality 64, and a sequential prior $p(\mathbf{z}_t|\mathbf{z}_{<t})$. It is similar to our background module with the same number of latent variables for encoder and decoder. As for the VRNN, we implement a VRNN with 128 dimensions of the LSTM hidden state and latent variable $\mathbf{z}$

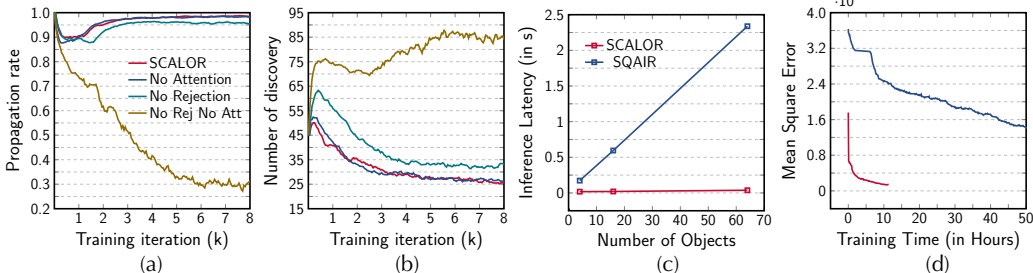

**Figure 8:** Attention-Rejection Ablation study and computational efficiency comparison. (a) Propagation rate, (b) Number of discovered objects, (c) Inference time and, (d) Training convergence time

and choose a convolutional network as the image encoder and decoder. The NLL value for our model is 28.30, and for the VAE and VRNN baseline it is 27.59 and 27.79 respectively. While learning a highly structured representation, SCALOR can still obtain a comparable generation quality.

### 5.3 Ablation Study and Computational Efficiency Comparison

**Ablation Study.** We perform an ablation study on the rejection mechanism and the propagation attention mechanism. Here, we train our model on the moving dSprites dataset where the number of objects varies from 6 to 36. We use two metrics to evaluate different architectures. The first metric is the propagation rate, which measures how long an object is propagated. Since, by design, objects always stay in a scene for more than one frame, the propagation rate can be regarded as a proxy to measure the success rate of tracking through propagation. The second metric counts the number of discoveries that occurred in the whole sequence. If the discovery module dominates, a high value will be observed. As we can see in Figure 8 (a), with no rejection (Rej) or attention (Att) in propagation, the propagation rate goes down to 30% in early training iterations and keeps decreasing as the training progresses. In this setting, we found that the model re-discovers objects in every frame without tracking them properly. This is also shown in Figure 8 (b), where it has a significantly larger number of discoveries. With the propagation attention mechanism, the propagation rate increases to 95%. This is because the attention mechanism reduces the cost of finding the matching object between time-steps in propagation. The model thus favors tracking in propagation over re-discovery. It also prevents the discovery module from propagation collapse. However, as we can see in Figure 8 (b), the re-discovery problem still exists in this setting. The rejection mechanism, however, makes the propagation rate converge to 1 even without propagation attention. The propagation attention together with the rejection produced more accurate boxes.

**Computational Efficiency.** In Figure 8 (c) and (d), we measure the inference time (c), and training convergence time (d). For measuring the inference time, we consider a hypothetical situation with $N$ objects in the first frame. The model is desired to discover and propagate them accordingly. Furthermore, we set the discovery capacity of SQAIR to be the same as the number of discovery grid cells in SCALOR. Figure 8(c) demonstrates how much time one forward step takes, as the number of objects/discovery capacity increases from 4 to 64. SQAIR's speed decreases linearly as both its discovery and propagation mechanisms process objects sequentially. In contrast, SCALOR does not suffer from such phenomena as discovery and propagation are done in parallel. Figure 8 (d) shows MSE convergence vs. training time when both models are trained on the MNIST VLD setting. SCALOR converges to a lower MSE than SQAIR and does so orders of magnitude faster.

## 6 Conclusion

We introduce SCALOR, a probabilistic generative world model aimed at visually modeling an environment of crowded dynamic objects. With the proposed parallel discovery-propagation and proposal-rejection mechanism, we improve the capacity of object density from a few objects to up to a hundred objects. Unlike previous models, the proposed model can also deal with dynamic backgrounds. These contributions consequently makes our model applicable, for the first time in this line of research, to natural scenes. An interesting future direction is to introduce more structure to the background and to model interactions among objects and the background.

### Acknowledgments

SA thanks Kakao Brain and Center for Super Intelligence (CSI) for their support.

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

# A  ALGORITHMS

---

**Algorithm 1:** Discovery Proposal-Rejection Inference

---

**Input:** $\mathbf{x}_{1:t}$ - image sequence up to current time-step
$\qquad \mathbf{M}_t^{\mathcal{P}}$ - propagated mask from current time-step
$\qquad \tau$ - rejection hyper-parameter

```
/* Feature encoding                                                     */
```
$\mathbf{e}_t^{\text{img}} = \text{ConvLSTM}(\mathbf{x}_{1:t})$
$\mathbf{e}_t^{\text{mask}} = \text{MaskEncoder}(\mathbf{M}_t^{\mathcal{P}})$
$\mathbf{e}_t^{\text{agg}} = \text{Concat}[\mathbf{e}_t^{\text{img}}, \mathbf{e}_t^{\text{mask}}]$
```
/* Proposal-Rejection (done in parallel)                               */
```
$\text{AcceptedList}_t = []$
**for** $h \leftarrow 1$ **to** $H$ **do**
$\quad$ **for** $w \leftarrow 1$ **to** $W$ **do**
$\qquad$
```
        /* Objects proposal                                           */
```
$\qquad \tilde{z}_{t,h,w}^{\text{pres}} \sim \text{Bern}(\cdot | f_{nn}^{\text{pres}}(\mathbf{e}_{t,h,w}^{\text{agg}}))$
$\qquad$ **if** $\tilde{z}_{t,h,w}^{pres} == 0$ **then**
$\qquad\quad |\quad$ **continue**
$\qquad$ **end**
$\qquad \tilde{\mathbf{z}}_{t,h,w}^{\text{where}} \sim \mathcal{N}(\cdot | f_{nn}^{\text{where}}(\mathbf{e}_{t,h,w}^{\text{agg}}))$
$\qquad \mathbf{g}_{\mathbf{t,h,w}}^{\text{att}} = \text{STN}(\mathbf{x_t}, \tilde{\mathbf{z}}_{\mathbf{t,h,w}}^{\text{where}})$  // Attended Glimpse
$\qquad \tilde{\mathbf{z}}_{t,h,w}^{\text{what}} \sim \mathcal{N}(\cdot | \text{GlimpseEnc}(\mathbf{g}_{\mathbf{t,h,w}}^{\text{att}}))$
$\qquad \mathbf{o}_{t,h,w}, \mathbf{m}_{t,h,w} = \text{STN}^{-1}(\text{GlimpseDec}(\tilde{\mathbf{z}}_{t,h,w}^{\text{what}}), \tilde{\mathbf{z}}_{t,h,w}^{\text{where}})$
$\qquad$
```
        /* Accept-Reject Test                                         */
```
$\qquad \delta = \frac{\mathcal{A}(\mathbf{m}_{t,h,w}^{\mathcal{P}} \cap \mathbf{M}_t^{\mathcal{P}})}{\mathcal{A}(\mathbf{m}_{t,h,w}^{\mathcal{P}})}$  // $\mathcal{A}$ - pixel area
$\qquad$ **if** $\delta < \tau$ **then**
$\qquad\quad |\quad \text{AcceptedList}_t.\text{Add}(\tilde{\mathbf{z}}_{t,h,w}^{\text{what}}, \tilde{\mathbf{z}}_{t,h,w}^{\text{where}}, \tilde{z}_{t,h,w}^{\text{pres}})$
$\qquad$ **end**
$\quad$ **end**
**end**

**Output:** $\text{AcceptedList}_t$

---

---

**Algorithm 2:** Propagation Inference

---

**Input:** $\mathbf{x}_{1:t}$ - image sequence up to current time-step

$\quad\quad$ PropList$_{t-1} = \{\mathbf{z}_{t-1,n}, \mathbf{h}_{t-1,n}\}_{n\in\mathcal{O}_{t-1}}$ - latent variable from previous time-step

```
/* Feature encoding                                                  */
```

$\mathbf{e}_t^{\text{img}} = \text{ConvLSTM}(\mathbf{x}_{1:t})$

```
/* Object tracking (done in parallel)                                */
```

PropList$_t$ = PropList$_{t-1}$

**for** $n \leftarrow 1$ **to** $\mathcal{O}_{t-1}$ **do**

$\quad$ $\mathbf{e}_{t,n}^{\text{att}} = f_{nn}^{\text{att}}(\text{STN}(\mathbf{e}_{t,n}^{\text{img}}, (\mathbf{z}_{t-1,n}^{\text{pos}}, \mathbf{z}_{t-1,n}^{\text{scale}})))$  // Feature map attention

$\quad$ $\mathbf{h}_{t,n} = \text{GRU}(f_{nn}([\mathbf{e}_{t,n}^{\text{att}}, \mathbf{z}_{t-1,n}^{\text{what}}, \mathbf{z}_{t-1,n}^{\text{pos}}, \mathbf{z}_{t-1,n}^{\text{scale}}, z_{t-1,n}^{\text{pres}}]), \mathbf{h}_{t-1,n})$

$\quad$ $\mathbf{e}_{t,n}^{\text{agg}} = f_{nn}^{\text{agg}}([\mathbf{e}_{t,n}^{\text{att}}, \mathbf{z}_{t-1,n}^{\text{what}}, \mathbf{z}_{t-1,n}^{\text{pos}}, \mathbf{z}_{t-1,n}^{\text{scale}}, \mathbf{h}_{t,n}])$

$\quad$ $\mathbf{z}_{t,n}^{\text{pos}}, \mathbf{z}_{t,n}^{\text{scale}} \sim \mathcal{N}(.|f_{nn}^{\text{where}}(\mathbf{e}_{t,n}^{\text{agg}}))$

$\quad$ $\mathbf{g}_{\mathbf{t},\mathbf{n}}^{\text{att}} = \text{STN}(\mathbf{x_t}, (\mathbf{z}_{\mathbf{t},\mathbf{n}}^{\text{pos}}, \mathbf{z}_{\mathbf{t},\mathbf{n}}^{\text{scale}}))$  // Attented Glimpse

$\quad$ $\mathbf{z}_{t,n}^{\text{what}} \sim \mathcal{N}(\cdot|\text{GlimpseEnc}(\mathbf{g}_{\mathbf{t},\mathbf{n}}^{\text{att}}))$

$\quad$ $z_{t,n}^{\text{depth}} \sim \mathcal{N}(\cdot|f_{nn}(\mathbf{e}_{t,n}^{\text{att}}, \mathbf{z}_{t,n}^{\text{what}}, \mathbf{h}_{t,n}))$

$\quad$ $z_{t,n}^{\text{pres}} \sim \text{Bern}(\cdot|f_{nn}^{\text{pres}}(\mathbf{z}_{t,n}^{\text{pos}}, \mathbf{z}_{t,n}^{\text{scale}}, \mathbf{z}_{t,n}^{\text{what}}, \mathbf{h}_{t,n}))$

$\quad$ $\mathbf{z}_{t,n}^{\text{where}} = (\mathbf{z}_{t,n}^{\text{pos}}, \mathbf{z}_{t,n}^{\text{scale}}, z_{t,n}^{\text{depth}})$

$\quad$ **if** $z_{t,n}^{pres} == 1$ **then**

$\quad\quad$ | PropList$_{t,n}$.Update($\mathbf{z}_{t,n}^{\text{what}}, \mathbf{z}_{t,n}^{\text{where}}, \mathbf{z}_{t,n}^{\text{pres}}$)

$\quad$ **else**

$\quad\quad$ | PropList$_{t,n}$.Delete()

$\quad$ **end**

**end**

**Output:** PropList$_t$

---

---

**Algorithm 3:** Background Module and Rendering

---

**Input:** $\mathbf{x}_t$ - image at current time-step,

$\quad\quad$ $\mathbf{o}_t = \{\mathbf{o}_{t,n}\}_{n\in\mathcal{O}_t}$ - object RGB glimpses

$\quad\quad$ $\mathbf{m}_t = \{\mathbf{m}_{t,n}\}_{n\in\mathcal{O}_t}$ - object masks glimpses,

$\quad\quad$ $\{(\mathbf{z}_{t,n}^{\text{pos}}, \mathbf{z}_{t,n}^{\text{scale}}), z_{t,n}^{\text{pres}}\}_{n\in\mathcal{O}_t}$ - object latents for position and presence

```
/* Foreground object rendering (done in parallel)                    */
```

**for** $n \leftarrow 1$ **to** $\mathcal{O}_t$ **do**

$\quad$ $\mathbf{x}_{t,n}^{\text{fg}} = \text{STN}^{-1}(\mathbf{o}_{t,n}, (\mathbf{z}_{t,n}^{\text{pos}}, \mathbf{z}_{t,n}^{\text{scale}}))$

$\quad$ $\boldsymbol{\gamma}_{t,n} = \text{STN}^{-1}(\mathbf{m}_{t,n} \cdot z_{t,n}^{\text{pres}} \sigma(-\mathbf{z}_{t,n}^{\text{depth}}), (\mathbf{z}_{t,n}^{\text{pos}}, \mathbf{z}_{t,n}^{\text{scale}}))$

$\quad$ $\boldsymbol{\gamma}_{t,n} = \text{normalize}(\boldsymbol{\gamma}_{t,n}, \forall n)$

**end**

$\mathbf{x}_t^{\text{fg}} = \sum_{n\in\mathcal{O}_t} \mathbf{x}_{t,n}^{\text{fg}} \boldsymbol{\gamma}_{t,n}$

```
/* Foreground mask rendering                                         */
```

**for** $n \leftarrow 1$ **to** $\mathcal{O}_t$ **do**

$\quad$ | $\mathbf{M}_{t,n} = \text{STN}^{-1}(\mathbf{m}_{t,n}, (\mathbf{z}_{t,n}^{\text{pos}}, \mathbf{z}_{t,n}^{\text{scale}}))$

**end**

$\mathbf{M}_t = \min(\sum_{n\in\mathcal{O}_t} \mathbf{M}_{t,n}, 1)$

```
/* Background rendering                                              */
```

$\mathbf{e}^{\text{bg}} = \text{BackgroundEncoder}(\text{Concat}[\mathbf{x}_t, (1 - \mathbf{M}_t)])$

$\mathbf{z}^{\text{bg}} \sim \mathcal{N}(.|f_{nn}^{\text{bg}}(\mathbf{e}^{\text{bg}}))$

$\mathbf{x}_t^{\text{bg}} = \text{BackgroundDecoder}(\mathbf{z}^{\text{bg}})$

```
/* Foreground background combination                                 */
```

$\mathbf{x}_t = \mathbf{x}_t^{\text{fg}} + (1 - \mathbf{M}_t) \odot \mathbf{x}_t^{\text{bg}}$

**Output:** $\mathbf{x}_t$

---

# B    ADDITIONAL QUANTITATIVE RESULT

Table 1 includes Multi Object Tracking Accuracy (MOTA) as well as precision-recall of the inferred bounding boxes (Bernardin & Stiefelhagen, 2008).

Table 2 provides a comparison of SCALOR to SQAIR and VRNN in terms of the reconstruction error (MSE) and negative log-likelihood (NLL). NLL is computed per pixel across the whole sequence.

| | Experimental Setting | MOTA ↑ | Precision ↑ | Recall ↑ | MAE ↓ |
|---|---|---|---|---|---|
| dSprites | SCALOR – VHD | 84.6% | 96.8% | 88.1% | 0.091 |
| | SCALOR – HD | 92.8% | 97.6% | 95.4% | 0.033 |
| | SCALOR – MD | 93.5% | 97.8% | 95.9% | 0.041 |
| | SCALOR – LD | 92.5% | 97.9% | 94.9% | 0.059 |
| | SCALOR – VLD | 93.8% | 96.3% | 98.9% | 0.067 |
| | SQAIR – LD | 20.2% | 97.6% | 64.1% | 0.335 |
| | SQAIR – VLD | 72.5% | 98.4% | 75.4% | 0.239 |
| MNIST | SCALOR – MD | 86.9% | 99.0% | 87.9% | 0.113 |
| | SCALOR – LD | 85.9% | 99.0% | 87.1% | 0.124 |
| | SCALOR – VLD | 94.8% | 97.7% | 98.8% | 0.040 |
| | SQAIR – VLD | 78.5% | 99.9% | 78.8% | 0.214 |
| | SCALOR – BG | 91.9% | 97.2% | 95.3% | 0.034 |
| | SCALOR – LG | 92.3% | 97.0% | 95.7% | 0.032 |

**Table 1:** Quantitative results of SCALOR for different experimental settings. "SCALOR - BG" refers to the dynamic background task for Experiment 3. "SCALOR – LG" refer to the length generalization experiment in Section C.4

| | | dSprites | | | | MNIST | | |
|---|---|---|---|---|---|---|---|---|
| | Method | VLD | LD | MD | HD | VLD | LD | MD |
| NLL | SCALOR | 22.16 | 22.41 | 22.56 | 22.88 | 7.45 | 7.54 | 7.82 |
| | VRNN | 22.33 | 22.80 | 23.28 | 23.30 | 7.62 | 7.70 | 7.71 |
| | SQAIR | 22.63 | 40.50 | - | - | 7.75 | - | - |
| MSE | SCALOR | 0.0010 | 0.0030 | 0.0024 | 0.0015 | 0.0029 | 0.0026 | 0.0010 |
| | VRNN | 0.0018 | 0.0071 | 0.0105 | 0.0190 | 0.0022 | 0.0054 | 0.0035 |
| | SQAIR | 0.0135 | 0.0918 | - | - | 0.0084 | - | - |

**Table 2:** Negative Log Likelihood and Mean Squared Error for SCALOR vs. baselines

## C    ADDITIONAL EXPERIMENTS ON DSPRITES AND MNIST

### C.1    FREQUENT DENSE DISCOVERY

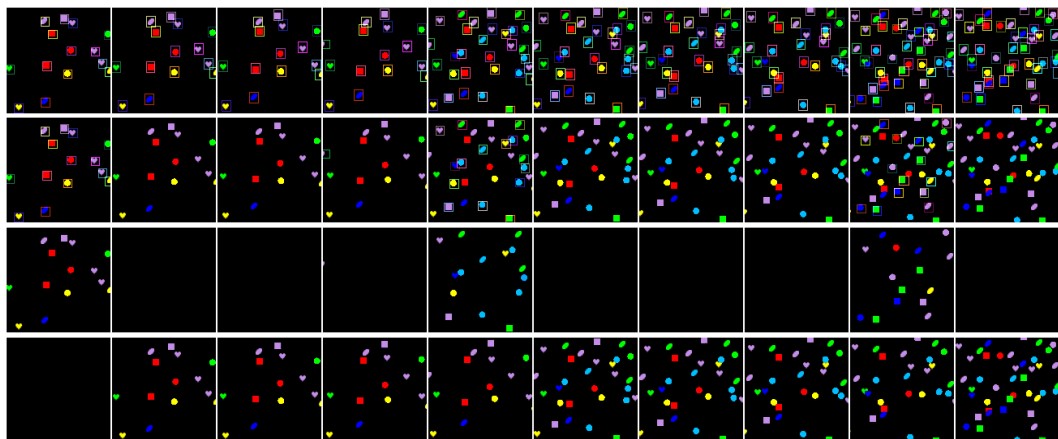

**Figure 9:** Frequent Dense Discovery experiment: a) First row: inferred bounding boxes superimposed on the original sequence. b) Second row: discovery bounding boxes. c) Third row: discovery reconstruction. d) Last row: propagation reconstruction

This experiment evaluates the ability to discover many newly introduced objects across time-steps. This is important because in many applications only key-frames of a video are available, i.e., frames at which significant changes happen. An example of a key-frame is a sudden change in the observer's viewpoint. This change of viewpoint introduces many new objects in the frame. Figure 9 shows one such instance. In this setting, 10–15 objects are introduced at the first, fifth, and ninth time-steps, respectively. As is shown, SCALOR is able to discover many new objects in each frame.

### C.2    VERY HIGH DENSITY

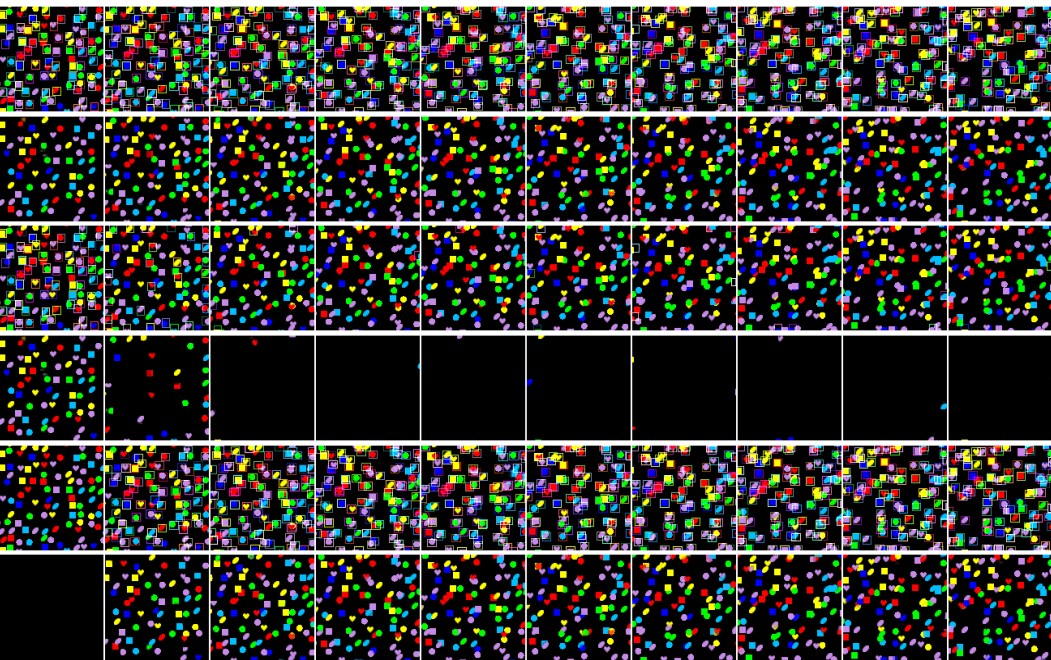

**Figure 10:** Sample from Very High Density setting: a) First row: inferred bounding boxes, b) Second row: overall reconstruction, c) Third row: discovery bounding boxes, d) Fourth row: discovery reconstruction, d) Fifth row: propagation bounding boxes, e) Sixth row: propagation reconstruction

Figure 10 shows samples from the Very High Density experiment. This experiment places 90–110 objects in the overall environment, around 90 of which are visible at every time-step on average. The discovery module contains $8 \times 8$ detection grid cells and thus can only detect up to 64 objects. Interestingly, as is shown, the model will identify as many objects as possible in the first time-step and discover the remaining in the second time-step. Furthermore, if too many objects are densely packed in one local region of the space, SCALOR will perform similarly detecting them through multiple frames.

## C.3 ABILITY TO HANDLE OVERLAP AND OCCLUSION

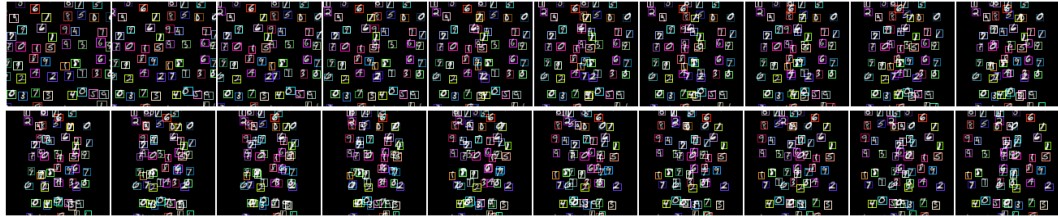

**Figure 11:** Highly occluded setting: Sample sequences in which objects move towards each other move more aggressively so overlap and occlusion happens more frequently

Figure 11 shows a sample of the MNIST settings where objects move towards each other more aggressively. In this setting, there is a higher chance of overlapping and occlusion. As shown, the identity of the objects is preserved when objects overlap with each other.

## C.4 GENERALIZATION TEST

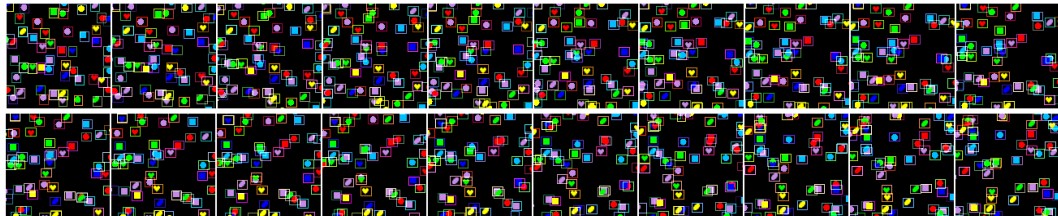

**Figure 12:** Generalization with respect to longer sequences. a) First row: bounding boxes for the first 10 time-steps. b) Second row: bounding boxes for the last 10 time-steps

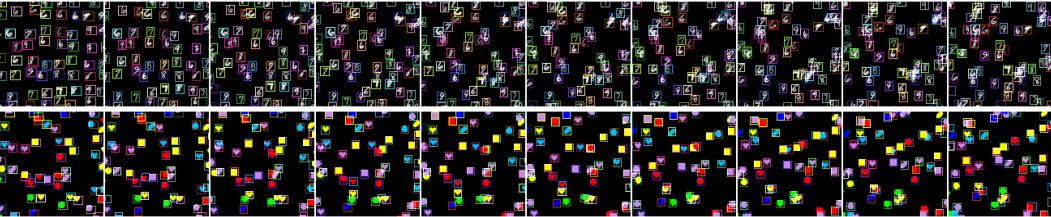

**Figure 13:** Generalization Experiment: a) First row: generalization to unseen shapes. b) Second row: generalization to a larger number of objects

We conduct three sets of experiments on generalization. In the first experiment, we investigate generalization to longer sequences. In this setting, the model is trained on trajectories of length 10 while being tested on trajectories of length 20. In the second experiment, we evaluate generalization in more crowded scenes. In this setting, the model is trained on 15–25 objects and tested on 50–60 objects. The third experiment tests the generalization of the model to unseen objects. The model is trained only on moving MNIST images containing digits 0 to 5 while being tested on images containing digits 6 to 9. Figures 12 and 13 show samples from these experiments. "SCALOR – LG" in Table 1 also provides tracking results for the "length generalization" setting.

## D EXPERIMENT DETAIL AND ADDITIONAL QUALITATIVE RESULTS ON GRAND CENTRAL STATION DATASET

For natural-scene experiments, we spatially split the video into 8 parts and create a dataset of 400k frames in total. We choose the first 360k frames for training and 40k frames for testing. Since the movement of pedestrians is too slow under 25 fps in the original video, we treat every other 7 frames as two consecutive frames in the dataset sequence. The length of the input sequence is 10, and each image is resized to $128 \times 128$.

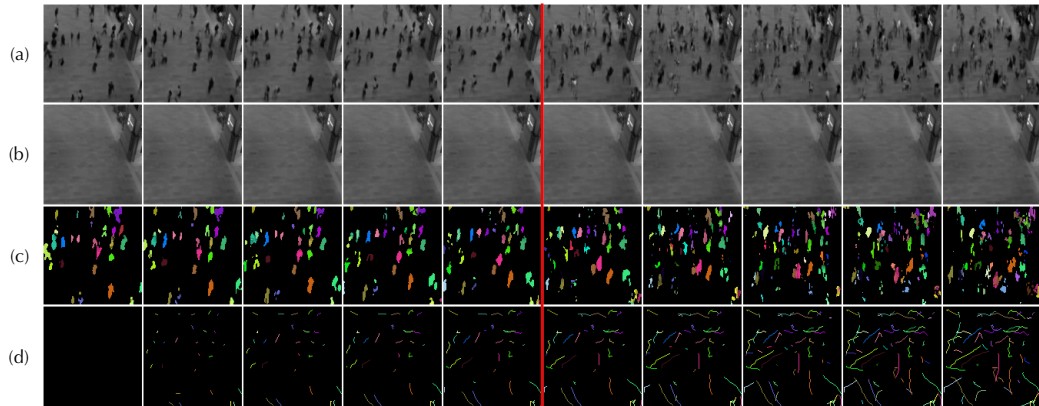

**Figure 14:** Conditional generation on Grand Central Station Dataset. The first 5 frames are observed, the next 5 frames are generated (after the red line). (a) Overall reconstruction and generation, (b) Conditional generation of background, (c) Conditional generation of segmented objects, (d) Conditional generation of movement trajectories

The generation result mentioned in Section 5.2 is provided in Figure 14. We provide additional qualitative results in Figures 15 to 20. Rows from top to bottom represent input sequence, overall reconstruction, extracted background, foreground segmentation mask with IDs, center position from each $\mathbf{z}_{t,n}^{\text{pos}}$ latent, and extracted trajectories based on the transition of center position. Additionally, Figures 18 to 20 show examples of conditional generation, where images starting from time-step 6 (after the red line) are generated sequences. Rows from top to bottom are conditional generations of the overall images, conditional generations of background, extracted conditional generation of each segmentation mask, and conditional generation of trajectories.

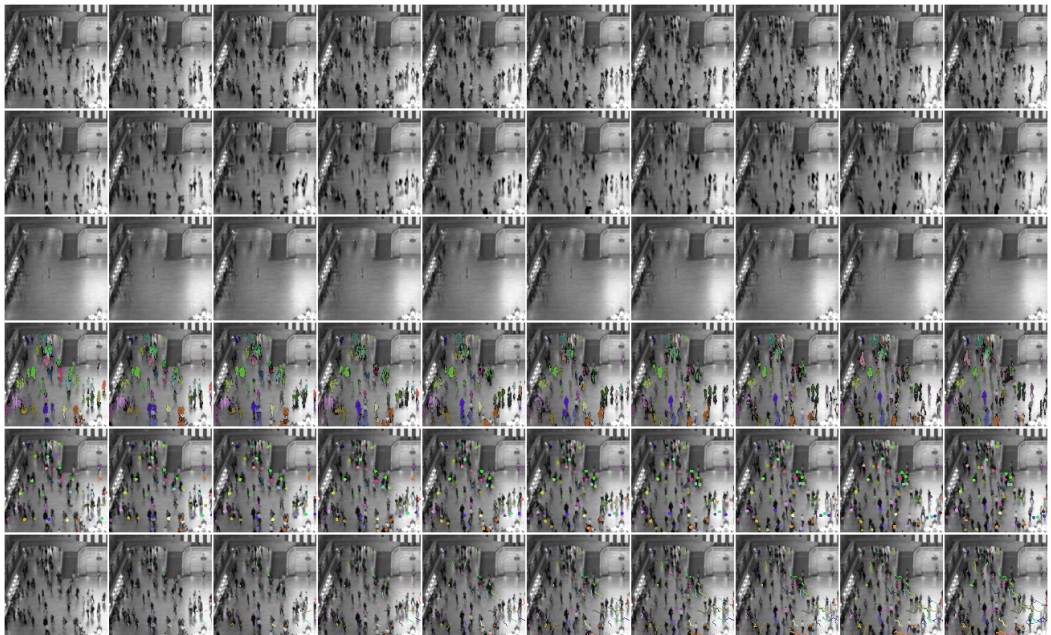

**Figure 15:** Tracking sample 1

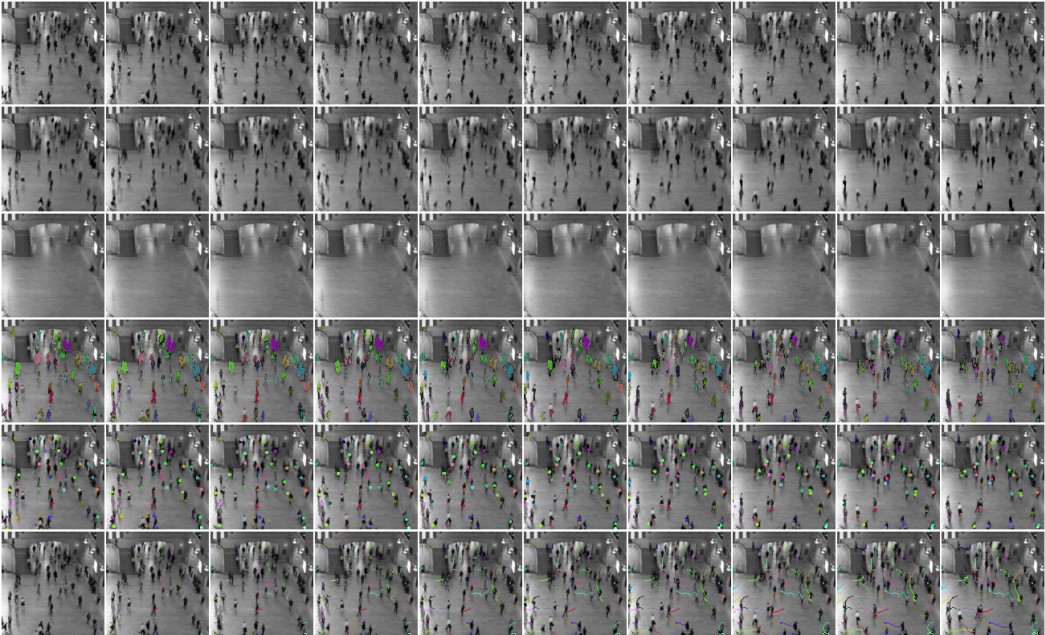

**Figure 16:** Tracking sample 2

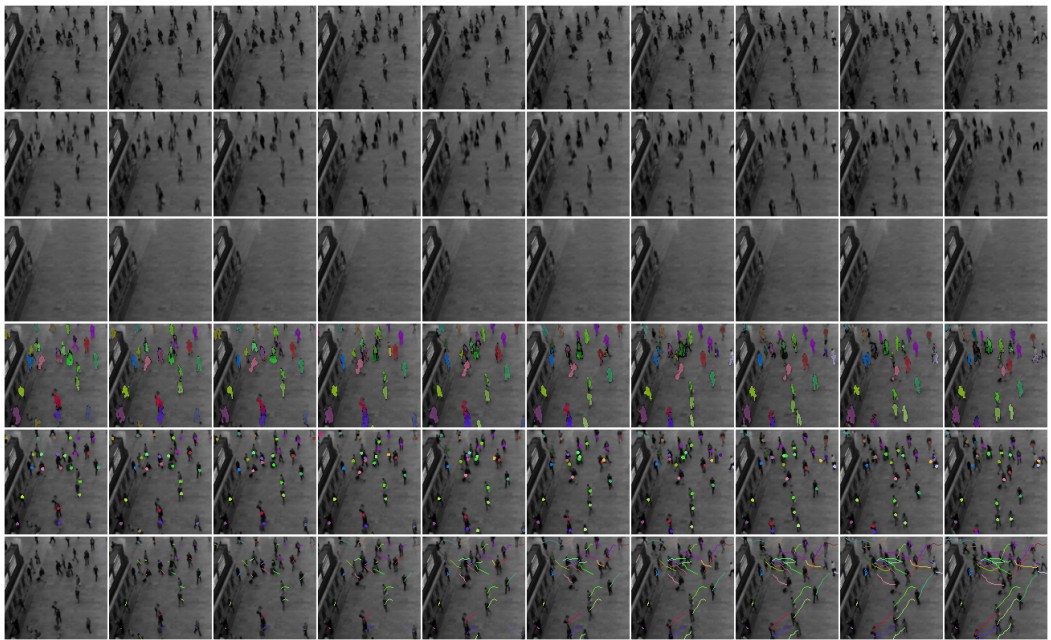

**Figure 17:** Tracking sample 3

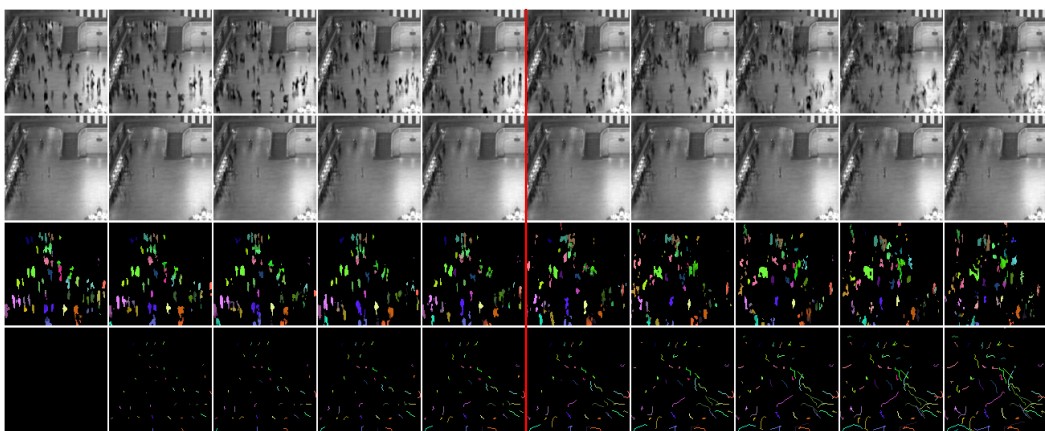

**Figure 18:** Conditional generation sample 1

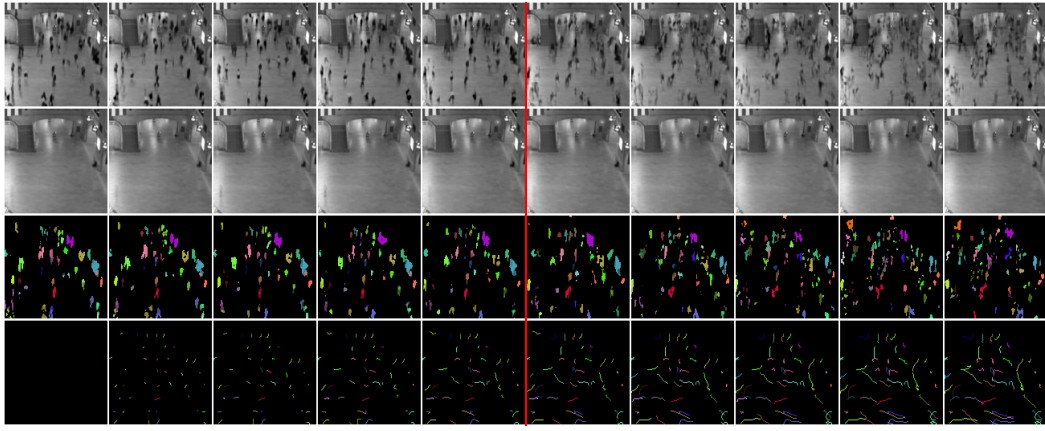

**Figure 19:** Conditional generation sample 2

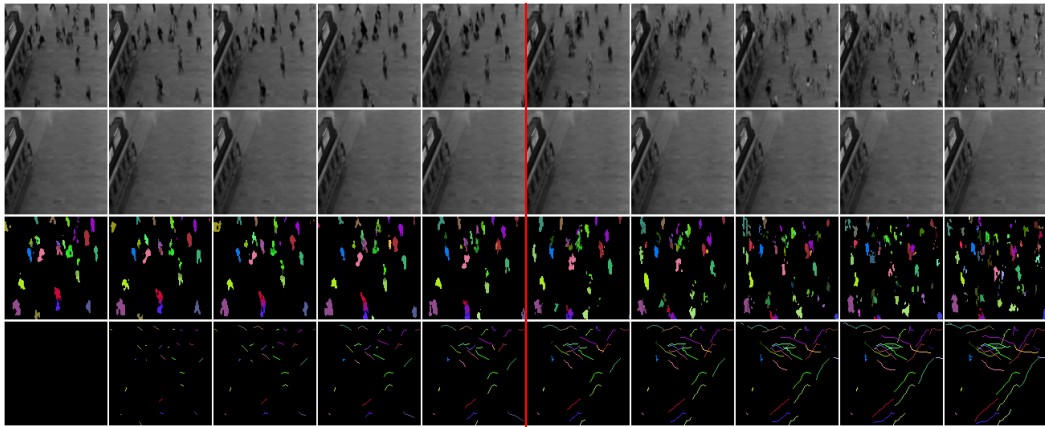

**Figure 20:** Conditional generation sample 3

## E    MODEL ARCHITECTURE DETAILS

In this section, we provide additional details of the architecture and hyperparameters used for pedestrian detection. When a new frame is provided, the network uses a fully convolutional encoder to obtain a $H \times W$ feature map. The feature map is fed into a convolutional LSTM to model the sequential information along the sequence. The convolutional LSTM is shared across discovery module and propagation module. The discovery module and propagation module also share the $\mathbf{z}^{\text{what}}$ encoder and decoder. The encoder has a convolutional network followed by one fully connected layer, while the glimpse decoder uses a fully convolutional network with sub-pixel layer (Shi et al., 2016) for upsampling. The background module shares a similar structure with the $\mathbf{z}^{\text{what}}$ encoder and decoder. It takes a 4-dimensional input, i.e., RGB and foreground mask, and outputs a 3-dimensional image. We use GRUs in propagation trackers and in prior transition networks.

We choose a batch size of 20 for the natural scene experiments and a batch size of 16 for MNIST/dSprites experiments. The learning rate is fixed at 4e-5 for natural image experiments and 5e-4 for dSprites/MNIST experiments. We use RMSprop for optimization during training. The standard deviation of the image distribution is chosen to be 0.1 for natural experiments and 0.2 for toy experiments. The prior for all Gaussian posteriors is set to standard normal. For the pedestrian tracking dataset, we constrain the range of $\mathbf{z}^{\text{scale}}$ so that the inferred width can vary from 5.2 pixels to 11.7 pixels, and the height can vary from 12.0 to 28.8, and both with a prior of the middle value in discovery. Similarly, we constrain $\mathbf{z}^{\text{scale}}$ on synthetic datasets so that it can vary from half to 1.5 times the actual object size. The $\mathbf{z}^{\text{pos}}$ variable in the propagation phase is modeled as the deviation of the position from the previous time-step instead of the global coordinate. The prior for $z^{\text{pres}}$ in discovery is set to be 0.1 at the beginning of training and to quickly anneal to 1e-3 for natural image experiments and 1e-4 for dSprites/MNIST experiments. The temperature used for modelling $z^{\text{pres}}$ is set to be 1.0 at the beginning and anneal to 0.3 after 20k iterations.

Full details of the architecture will be released along with our code.

