# OpenReview forum: "SCALOR: Generative World Models with Scalable Object Representations"
_ICLR.cc/2020/Conference — Accept (Poster)_

### Official Review · AnonReviewer2 · 2019-10-21
**Official Blind Review #2**

**Rating:** 6

**Review:**

UPDATE: My original main concern was the lack of baseline, but during the rebuttal period the authors have conducted the request comparison and addressed my questions satisfactorily. Therefore, I would recommend the paper be accepted.

---
Summary: This paper proposes a generative model and inference algorithm for discovering and propagating object latents in a way that scales to hundreds of objects. The key components of their approach is the parallel discovery and propagation of object latents as well as the explicit modeling of the background. The authors show that the model is able to model and generalize to scenes with hundreds of objects, including a real-world scene of a train station.

Research Problem: This paper tackles the problem of scaling object-oriented generative modeling of scenes to scenes with a large number of objects.

The main weakness of the submission is the lack of a baseline, and mainly for this result I would recommend rejecting the submission at its current state. However, if the authors are able to revise the submission to include such comparisons (with Kosiorek et al. (2018), van Steenkiste et al. (2018), and Alahi et al. (2016), detailed below), then I would highly consider accepting the paper, as the paper makes a novel contribution to modeling scenes with many more objects than previous work, as far as I am aware.

Strengths:
- The authors show that the method can model various synthetic and real-world datasets that show the efficacy of the method
- The method can also generalize to more objects and longer timesteps than trained on.

Weaknesses:
- The main weakness of the submission is the lack of a baseline. It would be important to understand the differences between Kosiorek et al. (2018), which the authors claim is the closest work to theirs, and van Steenkiste et al (2018), which also models objects in a parallel fashion. Alah et al. (2016) also takes an approach of dividing the scene into grid cells and also demonstrate results on modeling human trajectories.
- Motivation: Whereas the authors motivate the benefits for modeling objects, the motivation for specifically scaling to model hundreds of objects is less clear. It would be helpful for the authors to provide examples or arguments for the benefits of modeling so many objects at once. One argument against such a need is that humans only pay attention to a few number of objects at a time and do not explicitly model every possible object in parallel. One argument in favor of such a need is the ability to gain superhuman performance on tasks that could benefit from modeling multiple entities, such as playing Starcraft, detecting anamolies in medical scans, or modeling large scale weather patterns.
- A possible limitation of the method may be in modeling interactions between entities. What mechanism in the propagation step allows for modeling such interactions, and if not, how could such a mechanism be incorporated?
- How would SCALOR behave if the grid cells were smaller than the objects? In this case an object may occupy multiple grid cells. Would the authors provide an experiment analyzing this case? Would SCALOR model a object as multiple entities in this case (because the object spans multiple grid cells), or would SCALOR model the object with a single latent variable?

Question:
- What is the fundamental reason for why the structure of such a generative model would cause the latents to model objects, rather than something else, such as the image patches that show the empty space between objects?

Alahi, A., Goel, K., Ramanathan, V., Robicquet, A., Fei-Fei, L., & Savarese, S. (2016). Social lstm: Human trajectory prediction in crowded spaces. In Proceedings of the IEEE conference on computer vision and pattern recognition (pp. 961-971).

Van Steenkiste, S., Chang, M., Greff, K., & Schmidhuber, J. (2018). Relational neural expectation maximization: Unsupervised discovery of objects and their interactions. arXiv preprint arXiv:1802.10353.

Kosiorek, A., Bewley, A., & Posner, I. (2017). Hierarchical attentive recurrent tracking. In Advances in Neural Information Processing Systems (pp. 3053-3061).

**Experience Assessment:**

I have published one or two papers in this area.

**Review Assessment: Checking Correctness Of Derivations And Theory:**

I assessed the sensibility of the derivations and theory.

**Review Assessment: Checking Correctness Of Experiments:**

I assessed the sensibility of the experiments.

**Review Assessment: Thoroughness In Paper Reading:**

I read the paper thoroughly.

---

> ### Author Response · Authors · 2019-11-15
> **Response to Blind Review #2 (2/2)**
>
> ** Motivation
>
> Thanks for pointing this out. We agree with the suggestion, and will provide a clearer presentation of the motivation in our revision. We believe that the reviewer agrees with our perspective that it is not unusual to encounter natural scenes with many (tens or hundreds of) objects. The question is whether we need to explicitly attend to all of these or not. Regarding this, our perspective aligns with the reviewer’s argument that it is required for superhuman performance. Although we also agree with the limitation of the human attention capacity, we do not see a reason why we ought to artificially impose such a limitation in an AI system. For example, contemporary self-driving technology heavily takes advantage of this superhuman level of attention/detection capacity, although those approaches are based on expensive supervised learning systems. As another example, due to the limited capacity of human attention, we as humans need to resort to (sequential) search or scanning to find something among many objects, which is clearly more time-consuming than a parallel search. Also, considering that the architecture of modern computers is optimized for parallel processing, we believe that a contemporary AI system should maximize the utility of parallel processing. The parallel attention on the possible strategies used in the Monte-Carlo Search of the AlphaGo system is another strong evidence supporting this (human Go-players are limited in this ability.) We are grateful for the provided arguments on human attention and other examples. This is an interesting point and we will include it in our revision.
>
> ** Interactions between entities
>
> We agree that modeling interaction would indeed lead to a more comprehensive model. The main focus of this paper, however, is to make the current state-of-the-art (SQAIR) more scalable beyond just 3-4 objects. This focus on scalability, while not considering interaction, seems fairly reasonable, given that SQAIR also does not possess any interaction modeling (note that SQAIR has a relation module in its architecture but it is not for modeling interactions but rather for more accurate tracking, and their experiments do not show interaction.). Indeed, adding interaction modeling, e.g,. using a graph network of the trackers, will be a key point for future work following the present work.
>
> ** What if the grid cells were smaller than the objects
>
> Because the encoder is a deep CNN, the input receptive field corresponding to a grid cell is actually a quite large area of the input image. The encoder learns which part of the input image a grid cell should represent, and thus, it is actually not a problem. The proposed method will still model an object entirely when the object can occupy more than one cell. In fact, in our Low Density (LD) experiment setting and the additional Very Low Density (VLD) setting that will be added in the revised version, the object size (24*24) is larger than the cell size (8*8).
>
> ** About the fundamental reason the proposed method models the object and not the empty spaces
>
> One of the most fundamental reasons is that the model is inclined to predict as few bounding boxes as possible in the image. As we can see in the paper, in both the generation process and the inference process, we use 𝐳𝑝𝑟𝑒𝑠 to control whether we will generate the appearance latent variable 𝐳𝑤ℎ𝑎𝑡 and the location latent variable $\mathbf{𝐳}^\text{𝑤ℎ𝑒𝑟𝑒}$. If $\mathbf{𝐳}^\text{pres} = 1$, we introduce new KL terms for $\mathbf{𝐳}^\text{what}$ and $\mathbf{𝐳}^\text{𝑤ℎ𝑒𝑟𝑒}$ in the ELBO. This is equivalent to increasing the loss in the objective function. Moreover, to penalize unnecessary bounding boxes, we also assume a low probability of object existence in the prior distinction in the ELBO to further encourage fewer boxes. Therefore, to reduce the KL loss while maintaining a good reconstruction, the model converges to a behavior that learns the object representations properly.

---

> ### Author Response · Authors · 2019-11-15
> **Response to Blind Review #2 (1/2)**
>
> ** Baseline: SQAIR
>
> > Regarding the issue of baselines, please first refer to the relevant answer to reviewer 1.
>
> ** Baseline: Social LSTM, RNEM, and HART
>
> Thank you for pointing out these approaches. We notice that the reviewer mentions Kosiorek et al. (2018) in the main text but refers to Hierarchical Attentive Recurrent Tracking (HART, 2017) in the reference. We believe that the reference to HART is a mistake and the reviewer intended to point to SQAIR. (But we still include HART in the following discussion.) The four methods mentioned in the review are SQAIR, Social LSTM, RNEM, and HART. For the most related work SQAIR, please refer to the relevant answer to reviewer 1.
>
> For the other three approaches, we decided not to include them in our experiment for the following reasons. First, all three methods are deterministic (no uncertainty), while SCALOR and SQAIR are probabilistic latent variable models. Second, both Social LSTM and HART are supervised methods, while SCALOR and SQAIR are unsupervised. Social LSTM requires object coordinates as input instead of raw images, while SQAIR and SPAIR use only raw images as input. Thus, it is not practical to compare an unsupervised model to supervised models.
>
> RNEM [1], while unsupervised, is a deterministic model targeting a different task than our work. In particular, methods such as RNEM, IODINE [2], MoNet [3], and GENESIS [4] are scene decomposition methods constructing a scene via mixture components. SCALOR, SQAIR, SPAIR [5] and AIR [6] take a different approach based on attention (using bounding boxes) to learn “object representations” rather than a “scene decomposition”. Although they are relevant, these are actually quite different approaches. For example, the scene decomposition methods do not provide any explicit positions of objects, do not provide its bounding box, and do not explicitly provide counts of objects, or object-level appearance representations (but it is a scene decomposition level). Thus, we cannot measure tracking metrics. In contrast, the “object-oriented” methods normally cannot cope with full scenes with backgrounds, but only focus on spatially local objects – SCALOR is the first model among the object-oriented methods that can deal with backgrounds. Due to this significant difference between these two tasks, no paper has compared both lines of work as baselines. Rather, the comparison is made within the relevant line of work. For example, IODINE is compared to RNEM but not to AIR. NEM [7] also is not compared to AIR. Similarly, SPAIR is compared against AIR but not to NEM or RNEM. Therefore, it seems that a comparison to SQAIR and VRNN (for generation quality measured by NLL) are the only appropriate baselines focusing on the same task setting. In the revision, we thus focus on comparing our method to SQAIR and VRNN.
>
> [1] Van Steenkiste, S., Chang, M., Greff, K., & Schmidhuber, J. (2018). Relational neural expectation maximization: Unsupervised discovery of objects and their interactions. arXiv preprint arXiv:1802.10353.
>
> [2] Greff, K., Kaufmann, R. L., Kabra, R., Watters, N., Burgess, C., Zoran, D., ... & Lerchner, A. (2019). Multi-object representation learning with iterative variational inference. arXiv preprint arXiv:1903.00450.
>
> [3] Burgess, C. P., Matthey, L., Watters, N., Kabra, R., Higgins, I., Botvinick, M., & Lerchner, A. (2019). Monet: Unsupervised scene decomposition and representation. arXiv preprint arXiv:1901.11390.
>
> [4] Engelcke, M., Kosiorek, A. R., Jones, O. P., & Posner, I. (2019). GENESIS: Generative Scene Inference and Sampling with Object-Centric Latent Representations. arXiv preprint arXiv:1907.13052.
>
> [5] Crawford, E., & Pineau, J. (2019). Spatially invariant unsupervised object detection with convolutional neural networks. In Proceedings of AAAI.
>
> [6] Eslami, S. A., Heess, N., Weber, T., Tassa, Y., Szepesvari, D., & Hinton, G. E. (2016). Attend, infer, repeat: Fast scene understanding with generative models. In Advances in Neural Information Processing Systems (pp. 3225-3233).
>
> [7] Greff, K., van Steenkiste, S., & Schmidhuber, J. (2017). Neural expectation maximization. In Advances in Neural Information Processing Systems (pp. 6691-6701).

---

### Official Review · AnonReviewer1 · 2019-10-23
**Official Blind Review #1**

**Rating:** 6

**Review:**

I thank the authors for the detailed rebuttal, as well as for the updates to the text and several new experiments in the revised version of the paper. Most of my comments are addressed well. I am happy to improve my rating and recommend to accept the paper.

---

The paper proposes an approach for unsupervised detection and tracking of objects in videos. The method continues the "Attend, Infer, Repear" (AIR) and Sequential AIR (SQAIR) line of work, but improves on these previous approaches in terms of scalability and can thus be applied to scenes with tens of objects. The scalability is obtained by replacing, wherever possible, sequential processing of objects by parallel processing. Experiments are performed on three datasets: moving DSprites, moving MNIST, and the real-world "Crowded Grand Central Station" dataset. The method seems to work in all cases, as confirmed by quantitative evaluation on the first two datasets and a qualitative evaluation on all three.

I recommend rejecting the paper in its current state. On one hand, the results look quite good, and the method seems to indeed scale well to many objects. On the other hand, novelty is limited and the experiments are limited in that there are no comparisons with relevant baselines and no clear experiments showing the specific impact of the architectural modifications proposed in this paper. Moreover, the paper is over 9 pages, which, as far as I remember, requires the reviewers apply "higher standards". Overall, I would like the authors to comment on my concerns (below) and may reconsider my rating after that.

Pros:
1) Relatively clear presentation.
2) Judging from the extensive qualitative results and the (limited) quantitative evaluation, the method seems to work.
3) I appreciate the additoinal results on generalization and parallel disovery in the appendix.

Cons:
1) Novelty of the work seems quite limited. The main contribution is in improving the efficiency of object detection/tracking by parallelizing the computation, without much conceptual innovation. This might be a sufficient contribution (in the end, efficiency is very important for actually applying methods in practice), but then a thorough evaluation of the method would be expected (see further comments about it further). Moreover, the previously published SPAIR method by Crawford and Pineau seems very relevant and related, but is not compared against and is only briefly commented upon, despite the code for that method seems to be available online. I would like the authors to clarify the relation to SPAIR and preferably provide an experimental comparison.

2) The experiments are restricted. While there are quite many qualitative results, several issues remain:
2a) No baseline results are reported. It is thus impossible to judge if the method indeed improves upon related prior works. In particular, comparisons to SQAIR and SPAIR would be very useful. If possible, it would be useful to provide even more baselines, for instance some traditional tracking methods. Comparisons can be both in terms of tracking/reconstruction performanc, as well as in terms of computational efficiency. Both can be measured as functions of the number of objects in the scene.
2b) There are few quantitative resutls. In experiments 2-4 of section 5.1 it seems it would be fairly easy to introduce some, in particular, one could compare how do these more difficult settings compare to the "default" one. In section 5.2 given that the ground truth tracks are not available, evaluating tracking is challenging, but one could still compare NLL/reconstruction with appropriate baselines. The provided comparison to a vanilla VAE in therms of NLL is actually somewhat confusing - not sure what it tells the reader; moreover, I would actually expect the NLL of the proposed structured model to be better. Why is it not?
2c) Since the paper is largely about tracking objects through videos, it would be very usefuly to include a supplementary video with qualitative results.

3) (minor) Some issues with the presentation:
3a) I found the method description at times confusing and incomplete. For instance, it is quite unclear which exactly architectures are used for different components. The details can perhaps bee looked up in the SQAIR paper, but it would still be useful to summarize most crucial points in the supplementary material to make the paper more self-contained.
3b) The use of \citet vs \citep is often incorrect. \citet should be used when the author names are a part of the text, while \ciptp if the paper is cited in passing, for instance: "Smith et al. (2010) have shown that snow is white." vs "It is known that snow is white (Smith et al. 2010)."
3c) Calling AIR a "seminal work in the field of object detection" is not quite correct - object detection is a well-established task in computer vision, and AIR is not really considered a seminal work in that field. It is a great paper, but not really in object detection.

**Experience Assessment:**

I have published one or two papers in this area.

**Review Assessment: Checking Correctness Of Derivations And Theory:**

I assessed the sensibility of the derivations and theory.

**Review Assessment: Checking Correctness Of Experiments:**

I carefully checked the experiments.

**Review Assessment: Thoroughness In Paper Reading:**

I read the paper at least twice and used my best judgement in assessing the paper.

---

> ### Author Response · Authors · 2019-11-15
> **Response to Blind Review #1 (3/3)**
>
> ** Comparison to traditional tracking methods and is SCALOR (and SQAIR) a tracking model?
>
> Other similar works such as SQAIR do not conduct such a comparison because existing tracking methods are either supervised, not probabilistic, or cannot learn to render. Our contribution is not in the space of supervised tracking or non-generative modeling. For these reasons, we believe that SQAIR and VRNN (for generation quality) should be the proper baselines to compare to. However, we will make sure to better acknowledge previous work on tracking.
>
> ** How do these more difficult settings (in experiments 2-4 of section 5.1) compare to the “default” one.
>
> This is a good point. Thanks for pointing this out. We will provide a comparison to our default settings.
>
> ** In sec 5.2., given that the ground truth tracks are not available, evaluating tracking is challenging, but one could still compare NLL/recon with appropriate baselines.
>
> We will provide a comparison to VRNN in our revision. Note that SQAIR cannot be used because it does not work for that number of objects and is unable to cope with the background rendering. Hence, it seems that the best one can do is to compare NLL/recon to VRNN. We hope that the reviewer understands the difficulty of this research due to the total failure of the baseline in high-density settings.
>
> ** Why compare to VAE & why our NLL is not better than VAE.
>
> The goal of the comparison is to show that our model achieves its main goal (learning object-oriented representations) without losing the generation quality. Thus, VAE, which does not need to model temporal information and object-level representations is the appropriate baseline to evaluate the generation quality. Regarding the fact that the NLL of SCALOR is not better than that of VAE, it is actually a common misconception to expect a better generation quality for a discrete representation learning model like ours. Although a discrete structure in neural networks provides many advantages such as interpretability and compositionality, it generally comes with some performance degradation because it significantly limits the model space and optimization performance, compared to continuous representations. So, it is noteworthy to devise a model that uses the power of discrete latent representations while still having generation quality comparable to continuous models.
>
> ** Supplementary Video
>
> > We have made a project website where you can find videos. Link: https://sites.google.com/view/scalor/home
>
> ** Minor Comments
>
> > Thanks for the comments. All of these comments make sense, and we will incorporate all of them in our revision.

---

> ### Author Response · Authors · 2019-11-15
> **Response to Blind Review #1 (2/3)**
>
> ** Baseline (Comparison to SQAIR)
>
> We entirely agree that SQAIR should be our baseline. Before discussing our update plan regarding the baselines, we first would like to emphasize the difficulty in making our main baseline SQAIR work. To researchers working on this problem, it is well-known that SQAIR is very unstable and difficult to train even for a few objects. Additionally, it is almost impossible to train it beyond a few objects. As evidence, in the SQAIR paper, the authors only evaluate up to 2 objects for MNIST and 3 objects for the DukeMTMC dataset, although it is obviously more interesting to test beyond this trivial number of objects. Moreover, there have been a few methods following the SQAIR framework [1, 2, 3], but none of them has ever reported any results beyond a few objects. Further, we have thus far assigned the task of reproducing SQAIR to 8 students, and none of them have ever succeeded in making it work beyond these “few objects” settings. Finally, via personal communication, the SPAIR authors also confirmed that SQAIR does not work beyond a few objects for them either. Considering all these observations, it seems fairly reasonable to conclude that SQAIR is extremely difficult or almost impossible to scale beyond a few objects. In our paper, we also provide a reason why SQAIR may suffer from this scalability problem, which can also be considered as a minor contribution.
>
> Nevertheless, we agree that it is reasonable to provide a comparison to SQAIR for settings where SQAIR can work, namely scenes with up to 3~4 objects. In our revision, we will add this comparison. Also, an additional computational efficiency comparison over SQAIR and the proposed method will also be included in the experiment section.
>
> Further, for all density settings, including those where SQAIR totally fails, we will also provide a comparison to VRNN in terms of the generation/reconstruction quality. Regarding this, we would also like to note that the tracking is not the sole purpose of generative models such as SQAIR and SCALOR. The rendering performance is also an important factor. The purpose of this experiment is to show that our model can learn an object-oriented sequential representation without impeding the generation quality. Note that in general, introducing such a discrete (object-oriented) representation comes with some performance degradation because it limits the model space and optimization, compared to continuous representations. Thus, our goal is not to be significantly better than VRNN. Rather, achieving a comparable level of performance ought to be a sufficient achievement, given that our model learns useful structured representations.
>
> [1] Stanić, A., & Schmidhuber, J. (2019). R-SQAIR: Relational Sequential Attend, Infer, Repeat. arXiv preprint arXiv:1910.05231.
>
> [2] Kossen, J., Stelzner, K., Hussing, M., Voelcker, C., & Kersting, K. (2019). Structured Object-Aware Physics Prediction for Video Modeling and Planning. arXiv preprint arXiv:1910.02425.
>
> [3] Akhundov, A., Soelch, M., Bayer, J., & van der Smagt, P. (2019). Variational Tracking and Prediction with Generative Disentangled State-Space Models. arXiv preprint arXiv:1910.06205.

---

> ### Author Response · Authors · 2019-11-15
> **Response to Blind Review #1 (1/3)**
>
> The review text states that “the method seems to work” based on the qualitative results and the limited quantitative evaluation. This raises a discussion about the contribution of this paper.
>
> Regarding the quantitative comparison, our response is two-fold.
>
> ** Novelty. R1 claims “The main contribution is in improving the efficiency of object detection/tracking by parallelizing the computation without much conceptual innovation.”
>
> Although at first glance, one might assume that the parallelization may be a simple adaptation, we are not simply implementing a parallelization of a sequential model whose parallelization is already straightforward. Instead, we *identify the specific reasons that enable a parallelization* and then actually *make it work* with our own new observations, investigations, analysis, and experiments. This is not a minor contribution because, in the SPAIR paper, the authors actually state that the sequential computation (within an image) is crucial to obtain the desired results. Thus, given this previous state of the art, it is not trivial to devise a parallel algorithm without substantial conceptual innovation. Specifically, through our analysis and investigation, we first observe that an efficient parallelization that does not degrade the results is actually feasible, contrary to what was previously believed. Our new findings include (1) that in (physical) spatial space, two objects cannot exist in the same position, and, hence, the relation and interference from other objects should not be severe; (2) that considering the bottom-up encoding conditioning on the input image, each object should know what is happening in its immediate surroundings and thus should not need to communicate; and (3) that in the temporal general setting, the past behavior (trajectory) of an object ought to provide strong signals for the inference of an object’s latent in the future time step. (We will clarify these points more thoroughly in the revision.) Based on this reasoning, questioning the conclusion in the SPAIR paper, we propose our novel parallelization approach and show empirically that our insights and reasoning are correct, as R1 agrees that “the method seems to work.” Importantly, the SPAIR authors also confirmed via personal communication that they also recently realized that parallelization without performance degradation is possible even if they didn’t know it when they published SPAIR. This confirms that our findings constitute new knowledge correcting a false narrative on an important problem.
>
> In any case, the principal contribution of this paper should be considered from the perspective of sequential modeling. As described in the paper, it is highly non-trivial to make a sequential approach scalable. This is mainly because of the problem of combining a set of propagated objects with a newly discovered set of objects. This bipartite matching problem in object-oriented sequential representation learning has not been noticed in the community before because SQAIR is fully sequential for processing these objects. We found that this is an important problem to deal with in order to scale up the model beyond the previous state-of-the-art of operating on just a few objects. To resolve this, we devise our proposal-and-rejection mechanism, which may be considered as a key contribution along with the identification of the problem itself. Furthermore, demonstrating the feasibility of scaling up such a model to nearly a hundred objects with dynamic background as well as a complex natural scenes should be another dimension of the contribution, considering that the previous state-of-the-art involved operating on a few MNIST digits.
>
> ** Comparison to SPAIR (“not compared against SPAIR”)
>
> Unlike SQAIR, SPAIR is not a temporal model. It only works on static images, not on video feeds. Hence, it cannot track objects, but would need to re-discover objects for each image without providing any tracking information. It does not deal with propagation and discovery (everything should be re-discovered). It does not deal with the background. Given all these reasons, despite our discovery mechanism being partly inspired by SPAIR, a comparison with SPAIR is not an obvious point for evaluation regarding our claimed contributions in the paper. Note that our main claim is that the parallel discovery combined with temporal propagation modeling should be better than SQAIR. We thus believe, as pointed out by R1, that a comparison to SQAIR is a more reasonable one, which we further explain below.

---

### Official Review · AnonReviewer3 · 2019-10-27
**Official Blind Review #3**

**Rating:** 6

**Review:**

This paper proposes a generative model for scalable sequential object-oriented representation. The paper proposes several improvements based on the method SQAIR (Kosiorek et al. 2018b), (1) modeling the background and foreground dynamics separately; (2) parallelizing the propagation-discovery process by introducing the propose-reject model which reducing the time complexity. Finally, the proposed model can deal with orders of magnitude more objects than previous methods, and can model more complex scenes with complex background.

Accept.
The paper is clearly written and the experimental results are well organized. The results in the paper may be useful for unsupervised multi-objects tracking .I have one concern here,
（1）As argued in the paper, previous methods are difficult to deal with the nearly a hundred objects situation and there is no direct comparison for these methods. So has the author compared the method SCALOR with previous methods in few objects setting? Does the technical improvements of the method benefit in the few objects setting?

**Experience Assessment:**

I have read many papers in this area.

**Review Assessment: Checking Correctness Of Derivations And Theory:**

I did not assess the derivations or theory.

**Review Assessment: Checking Correctness Of Experiments:**

I did not assess the experiments.

**Review Assessment: Thoroughness In Paper Reading:**

I made a quick assessment of this paper.

---

> ### Author Response · Authors · 2019-11-15
> **Response to Blind Review #3**
>
> Thank you for the suggestion. Yes, in the revision, we will add two additional experiments in a “Very Low Density (VLD)” setting, containing up to 4 objects, in which SQAIR works properly, and several different metrics to compare our method to SQAIR quantitatively.  As we show quantitatively, our method can outperform SQAIR even in those settings, leading to more accurate and consistent bounding boxes.

---

### Author Response · Authors · 2019-11-15
**For All Reviewers**

We thank all the reviewers for taking the time to read our paper and provide insightful feedback and suggestions. We will upload a new version of the paper. The main focus of the revision is to provide quantitative comparisons to baselines, which was the main concern about the paper. For this, we performed a significant amount of additional experiments to provide the following quantitative evaluation:

1) We add two additional “Very Low Density” experiment settings for Moving MNIST and Moving dSprites and introduce SQAIR as the baseline for comparison.
2) We also introduce VRNN as a baseline for all experimental settings to compare the reconstruction quality
3) An ablation study on the proposal-and-rejection mechanism is added to the experiment section.
4) We add an additional comparison experiment between SCALOR and SQAIR with respect to computational efficiency. Inference latency and training convergence time are used as metrics.

We also make the following general updates

5) To better illustrate the proposed architecture, we add an architecture diagram for the overall structure on the main text and also append pseudo-codes for each component in the appendix to explain each part in detail.

We have also created a project webpage with additional qualitative examples and video of SCALOR: https://sites.google.com/view/scalor/home

We will respond to each reviewer’s points in detail in the comments below. We believe we have addressed each reviewer’s concerns and look forward to hearing feedback about the updated version of our paper. We hope the reviewers can take our responses and revisions into consideration when evaluating our final score.

---

### Decision · Program_Chairs · 2019-12-19

**Decision:**

Accept (Poster)

**Comment:**

After the author response and paper revision, the reviewers all came to appreciate this paper and unanimously recommended it be accepted.  The paper makes a nice contribution to generative modelling of object-oriented representations with large numbers of objects.  The authors adequately addressed the main reviewer concerns with their detailed rebuttal and revision.